# EHRSQL: A Practical Text-to-SQL Benchmark for Electronic Health Records

**Gyubok Lee[1], Hyeonji Hwang[1], Seongsu Bae[1], Yeonsu Kwon[2],**
**Woncheol Shin[1], Seongjun Yang[1], Minjoon Seo[1], Jongyeup Kim†, Edward Choi[1]**
KAIST, Daejeon[1] Seoul Women's University, Seoul[2]
College of Medicine, Konyang University, Daejeon†
{gyubok.lee,localh,seongsu,swc1905,seongjunyang,
minjoon,edwardchoi}@kaist.ac.kr[1]
dustn1259@swu.ac.kr[2], jykim@kyuh.ac.kr†

## Abstract

We present a new text-to-SQL dataset for electronic health records (EHRs). The utterances were collected from 222 hospital staff, including physicians, nurses, insurance review and health records teams, and more. To construct the QA dataset on structured EHR data, we conducted a poll at a university hospital and templatized the responses to create seed questions. Then, we manually linked them to two open-source EHR databases—MIMIC-III and eICU—and included them with various time expressions and held-out unanswerable questions in the dataset, which were all collected from the poll. Our dataset poses a unique set of challenges: the model needs to 1) generate SQL queries that reflect a wide range of needs in the hospital, including simple retrieval and complex operations such as calculating survival rate, 2) understand various time expressions to answer time-sensitive questions in healthcare, and 3) distinguish whether a given question is answerable or unanswerable based on the prediction confidence. We believe our dataset, EHRSQL, could serve as a practical benchmark to develop and assess QA models on structured EHR data and take one step further towards bridging the gap between text-to-SQL research and its real-life deployment in healthcare. EHRSQL is available at `https://github.com/glee4810/EHRSQL`.

## 1 Introduction

Electronic health records (EHRs) are relational databases that store a patient's entire medical history in the hospital. From hospital admission to patient treatment and discharge, all medical events that happened in the hospital are recorded and stored in the EHR. The records cover a wide range of clinical knowledge, from individual-level information to group-level insight, in various forms, including tables, text, and images [16, 17, 11, 15]. As a vast and comprehensive knowledge base, hospital staff, including physicians, nurses, and administrators, constantly interact with EHRs to store and retrieve patient information to make better clinical decisions [32, 3].

Meanwhile, most hospital staff interact with the EHR database using pre-defined rule conversion systems. To look for information beyond the rules, one needs to undergo special training to modify and extend the system [32]. As a result, it is a massive bottleneck for the users to fully utilize the information stored in the EHR. An alternative way to tackle this problem is to build a system that can automatically translate questions directly into the corresponding SQL queries. Then, the users can simply type or verbally ask their questions to the system, and it will return the answers without going through any complicated process. Such systems will unleash the potential of EHR data and tremendously speed up any task involving data retrieval from the database.

36th Conference on Neural Information Processing Systems (NeurIPS 2022) Track on Datasets and Benchmarks.

Existing datasets that tackle question answering (QA) over structured EHR data are MIMICSQL [32] and emrKBQA [27]. MIMICSQL is the first dataset for healthcare QA on MIMIC-III [16], where the questions are automatically generated with pre-defined templates. emrKBQA is another dataset on MIMIC-III derived from emrQA [23], a clinical reading comprehension dataset. However, through a poll at a university hospital, we discover that the existing datasets are far from fulfilling the actual needs in the hospital workplace in several aspects.

We present EHRSQL[1], a new large-scale text-to-SQL dataset linked to two open-source EHR databases—MIMIC-III [16] and eICU [25]. To the best of our knowledge, our work is the first EHR QA dataset that reflects the diverse needs of hospital staff while introducing practical issues in deploying healthcare QA systems. The unique challenges posed by our dataset are threefold:

- **Wide range of questions:** We conducted a poll at a university hospital to collect questions frequently asked on structured EHR data. The total number of respondents was 222 people with varying years of experience in their professions (see Figure 1). After filtering and templatizing the responses, the resulting templates cover a wide range of questions, including retrieving patient records (*e.g.*, vital signs measured and hospital cost) and conducting complex group-level operations (*e.g.*, retrieving the top N medications prescribed after being diagnosed with a disease and calculating the N-year survival rate). See Table 1 for more examples.

- **Time-sensitive questions:** Based on the poll, we learned that real-world questions in the hospital workplace are rich in time expressions, as time is one of the most crucial aspects of healthcare. To reflect this in the dataset, we systematically categorized time into multiple expression types (*e.g.*, absolute, relative, and mixed), units (*e.g.*, hospital visit, month, and day), and interval types (*e.g.*, since, until, and in). Then, we combined the categorized time with the question templates to simulate time-sensitive questions asked in the hospital. More details are discussed in Section 3.1.2.

- **Trustworthy QA systems:** Developing trustworthy systems is crucial for the adoption of AI in healthcare. Likewise, QA systems need to return only accurate answers and refuse to answer questions beyond their capabilities. To test this, we include unanswerable questions in the dataset, utilizing the remaining utterances from the poll result. These utterances are unanswerable due to incompatibility with the database schema or requiring external domain knowledge.

## 2 Related Works

**EHR QA**    Pampari et al. [23] proposed emrQA for question answering on clinical notes based on physicians' frequently asked questions. Later, Raghavan et al. [27] proposed emrKBQA[2], which adapted emrQA to structured patient records in MIMIC-III. Due to the nature of the original dataset targeting outpatients, the scope of the questions in emrKBQA is limited to mostly asking about patients' test results [27]. Recently, Wang et al. [32] proposed MIMICSQL, which first tackles EHR QA with the text-to-SQL generation task. To construct the dataset, they automatically generated questions based on pre-defined rules and filtered them through crowd-sourcing. However, MIMICSQL contains a limited scope of questions and simple SQL queries restricted to five tables, which are far from the actual use case in the hospital [24]. EHRSQL, on the other hand, originates from the actual poll result and covers a variety of real-world questions (spanning 13.5 tables on average) frequently asked on structured EHR data (see Table 1).

**Semantic parsing datasets**    Semantic parsing has been one of the most active areas of research in natural language processing over the past few decades [13, 37, 34, 9]. It involves converting natural language utterances to logical form representations, which are often executable programs such as Python scripts and SQL queries. WikiSQL [40] and Spider [35] are leading datasets that assess semantic parsing models on unseen databases. WikiSQL provides a wide range of database domains extracted from HTML tables in Wikipedia, but it only contains simple SQL queries and single tables. Spider aims to tackle complex queries, including joining and set operations, on multiple tables in multiple database domains. Recently, more realistic datasets have been proposed to bridge the gap between academic and practical settings in semantic parsing. KaggleDBQA [19] utilizes real-world databases from Kaggle and constructs domain-specific questions without relying on the database

---

[1]The dataset is distributed under the CC BY-SA 4.0 license.

[2]emrKBQA is yet to be publicly released at the time of this work.

Table 1: Sample utterances in EHRSQL (SQL queries are labeled differently according to the MIMIC-III and eICU schemas).

| Type | | Sample question | Department |
|---|---|---|---|
| Question diversity | Demographics | Tell me the **birthdate** of patient 92721? | Nursing |
| | Prescription | When was the first **prescription** time of patient 20000? | Nursing |
| | Vital sign | What was the last **arterial bp [systolic]** for patient 23042? | Physician, Nursing, Health records, Other |
| | Cost | What is the **average total hospital cost** that involves non-invasive mech vent? | Insurance review |
| | Survival rate | What is the five diagnoses which have the lowest **four year survival rate**? | Nursing |
| | Longitudinal statistics | What are the five most commonly **prescribed drugs** for patients who have been **diagnosed with hypotension nos** earlier since 2104 within 2 months? | Physician, Nursing |
| Time-sensitive question | Absolute | Is the heart rate of patient 2518 measured **at 2105-12-31 12:00:00** less than the value measured **at 2105-12-31 09:00:00**? | Physician |
| | | Tell me the last specimen test given to patient 51177 in **10/2105?** | Nursing |
| | Relative | Tell me patient 6990's daily average gastric meds intake **on the last ICU visit**. | - |
| | Mixed | Tell me the medication that patient 3929 has been prescribed for the first time **in 05/this year**. | Nursing |
| Unanswerable question | Requiring external knowledge | What is the name of the **medication which should not be administered** during the contrast arteriogram-leg treatment? | Nursing |
| | Beyond DB schema | When is the **next earliest hospital visit** of patient 73652? | Nursing |

schema. SEDE [12] deals with complex SQL queries that Stack Exchange users ask in real life. Our motivation aligns with theirs in that additional challenges arise in real-world healthcare QA systems.

**Unanswerable questions in QA**    In the early days of question answering, unanswerable questions were generated via distant supervision [18, 4], rule-based editing [14], or adversarial creation by crowd workers [28]. Yet, in semantic parsing, most works assume that all input questions are valid and can be answered; however, this is not true in practice [38]. Retrieving answers to all the input questions is not always desirable for the model to ensure system reliability. To address this, Zhang et al. [38] utilized other text-to-SQL and chit-chat datasets to construct unanswerable questions and posed the problem as a classification task to detect unanswerable questions. Our work, on the other hand, addresses both system reliability and semantic parsing simultaneously and the unanswerable questions are naturally collected through the poll. In task-oriented dialog systems, the same task has been tackled in the name of Out-of-Domain (OOD) detection, and several recent works approach this problem without explicitly showing OOD samples to the model during training for practical reasons. Following their setup, our unanswerable questions are only included in the validation and test sets.

## 3   Dataset Construction

**Data collection**    The motivation behind our work is to construct a dataset that reflects the actual needs of hospital staff and tackles several practical issues that can arise in real-world healthcare QA systems (see Table 1). To this end, we collaborated with the Konyang University Hospital[3] and conducted a poll to collect real-world questions as if one is asking an AI speaker what they frequently look for in structured information in the EHR. In addition, to clarify what machines can and cannot answer, we provided typical negative examples of what should not be asked, such as requiring external knowledge, ambiguous or qualitative statements, or asking for reasons behind clinical decisions. As a result, we gathered a total of 1,742 utterances, and the number of valid respondents was 222 (see Figure 1 for respondent demographics).

### 3.1   Question and SQL Generation

#### 3.1.1   Question template

After the poll, we filtered out the utterances that did not meet the criteria, including ambiguous statements, those that require external knowledge, or those that go beyond the database schema. We

---

[3]https://www.kyuh.ac.kr/eng/

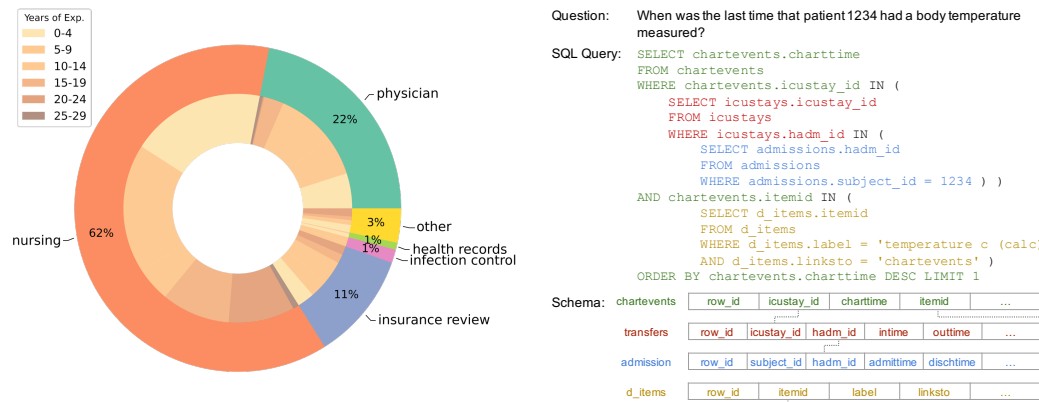

Figure 1: Demographics of the 222 respondents by hospital departments and the years of experience.

Figure 2: EHR schema is structured in a hierarchy, and real-world questions often require multi-step reasoning over tables.

then categorized the utterances into three patient-based scopes: a single patient, a group of patients, and no patient. For each scope, duplicate utterances, but differently phrased, were merged into a single question template. We also manually added question templates that were simple variants of existing templates. For example, we modified the type of *medical events* (*e.g.*, lab test, prescription, etc.) in question templates, such as from "Count the number of times that a patient had a lab test" to "Count the number of times that a patient received a prescription." To cover more questions on longitudinal statistics, we linked two different medical events together whenever possible. For example, the question "What are the top N prescriptions after a diagnosis," which was originally collected, is extended to "What are the top N lab tests after a diagnosis." Finally, the filtered-out utterances were also templatized and considered unanswerable. The unanswerable questions included side effects, next check-up schedules, primary care doctor, patient's personal information, whether a patient has signed a consent form, diagnosis received in other departments, and more.

Table 2: Sample question templates. The full list is reported in Supplementary B.1.

| Patient scope | Question template |
|---|---|
| None | What is the intake method of {drug_name}? |
| Individual | Has patient {patient_id} received any procedure [time_filter_global]? |
| Group | Count the number of hospital visits of patient {patient_id} [time_filter_global]. |
| | What are the top [n_rank] frequent procedures that patients received [time_filter_within] after having been diagnosed with {diagnosis_name} [time_filter_global]? |
| Unanswerable | What is the effect of {drug_name}? |

The resulting question templates are natural utterances with *slots* that are later filled with pre-defined values or database records (see Table 2). Slots containing "time_filter" are processed in the time template sampling stage, described in Section 3.1.2. We eliminate any ambiguity in the question templates that could cause the model to infer table or column names incorrectly. For example, "When did this patient get [slot] today?" is removed because it does not tell us whether we are interested in drugs, lab tests, or something else unless the slot is filled. Such linguistic ambiguity is later introduced in the paraphrase generation stage described in Section 3.3. As a result, we curated 230 question templates (174 for answerable and 56 for unanswerable) based on the MIMIC-III and eICU schemas. For answerable questions, their corresponding SQL queries were labeled for each database, which is further discussed in Section 3.1.3.

### 3.1.2 Time template

Since the healthcare domain is inextricably linked to time, many utterances we collected were rich in time expressions. To reflect this in the dataset, we developed three time filter types and assigned different time factors (expression types, units, and interval types) to each filter type to compose a single time template. First, we define a "global" time filter ([time_filter_global]) that

constrains the full range of time we are interested in. Second, within the global time filter, we can also use a "within" time filter (`[time_filter_within]`) to indicate the time range between two or more medical events. Third, we can point to an exact time (`[time_filter_exact]`), such as "last" measurement or "at 2105-12-31 09:00:00" if we know the exact time or the order of an event.

Each time filter type has three factors and an extra option: 1) *Expression type* is whether the filter is phrased in an absolute, relative, or mixed time expression (*e.g.*, last year, in 2022, etc.); 2) *Unit* is the granularity of time, such as a standardized unit (*e.g.*, year, month, day, etc.) or an arbitrary unit of events (*e.g.*, hospital visit, ICU visit); 3) *Interval type* specifies the type of time interval (*e.g.*, since, until, in, etc.). Depending on the use case, *Option* allows to choose one exact event (*e.g.*, first, last) among the filtered events. Table 3 shows sample time templates and their natural language (NL) *time expressions*. It is important to note that each time template has its corresponding NL time expression (*e.g.*, since {month}/{day}/{year}) and *SQL time pattern* (*e.g.*, WHERE strftime('%Y-%m-%d',[time_column]) >= '{year}-{month}-{day}'), while the question templates are originally created in natural language. The NL time expressions are combined with the question templates to form complete questions. The SQL time patterns are combined with the labeled SQL queries to form complete SQL queries (see Section 3.1.4 for details). The full list of time templates (NL and SQL pairs) is reported in Supplementary B.2.

Table 3: Sample time templates and their NL time expressions.

| Time filter type | Expression type | Unit | Interval type | Option | Time template | NL time expression |
|---|---|---|---|---|---|---|
| `[time_filter_global]` | absolute | day | since | - | abs-day-since | since {month}/{day}/{year} |
| | relative | year | in | last | rel-year-last | last year |
| | | | | this | rel-year-this | this year |
| `[time_filter_within]` | - | hospital | in | - | within-hosp | within the same hospital visit |
| | - | day | in | - | within-day | within the same day |
| `[time_filter_exact]` | relative | exact | at | - | exact-first | first |

### 3.1.3 SQL annotation

The SQL annotation process was conducted manually by four graduate students over the course of five months, with repeated revisions. Since the questions were collected independently of the database schema, SQL labeling required numerous assumptions (*e.g.*, which medical event is mapped to which column in the database, the choice of rank functions in SQL, the occasion of using DISTINCT, etc.). To sync template and value sampling, the same slots introduced in Section 3.1.1 and 3.1.2 (*e.g.*, {drug_name}, [n_rank], [time_filter_global]) were also used in SQL labeling as placeholders. The SQL queries for question and time templates were labeled by one person, followed by a reviewer for each database. Unlike most SQL datasets, the students were asked to avoid using JOIN, which is a go-to operation when retrieving information across multiple tables. In MIMIC-III and eICU, several tables contain more than 100 million rows (*e.g.*, 330 million rows in MIMIC-III chartevents) as the records are stored in a "log" based manner. In fact, it is extremely inefficient to merge all the tables in such large databases without understanding what specific columns are needed to answer a question. As a result, the students were asked to utilize the hierarchical structure of the EHR schema, as illustrated in Figure 2, and labeled each query in a nested manner whenever possible. SQL annotation details and the comparison between JOIN and nesting-based queries are discussed in Supplementary C.

### 3.1.4 Template combination and data generation

Data generation starts by selecting a question template, followed by three sampling stages that add semantic variety: *operation value sampling*, *time template sampling*, and *condition value sampling* (See Figure 3). Stage 1 involves sampling operation values from pre-defined values that are independent of the database schema or records (*e.g.*, maximum, five-year, two or more times, etc.). Then, time templates are sampled in Stage 2 based on the time filter types that each question template can have. Stage 2 in Figure 3 illustrates that the time templates (`[time_filter_global]` and `[time_filter_within]`) have already been sampled and converted into NL time expressions. More details are discussed in Supplementary B.3. Finally, condition value sampling (*e.g.*, {diagnosis_name}, {year}) occurs in Stage 3, and the slot filling process is complete. This

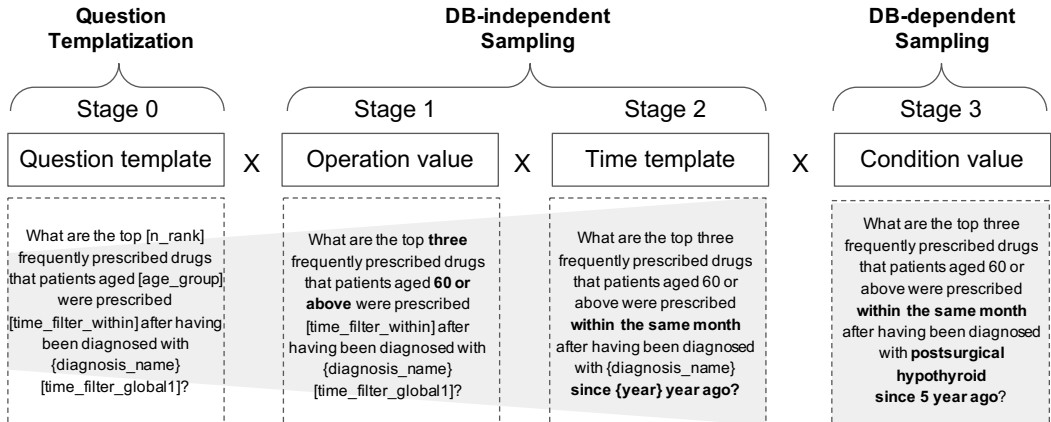

Figure 3: Question generation process. The shaded region refers to the expansion of semantic diversity in the template pool.

final stage is where the actual database records are sampled, and all generated SQL queries are checked to have at least one valid answer. Once the query is successfully executed, the pair of SQL and the corresponding question is added to the data pool, which is later used for data splitting.

## 3.2 Database Pre-processing

**Cost table** We modified the original MIMIC-III and eICU databases to include a cost table in order to reflect hospital administration and insurance-related questions. When constructing the table, we referred to the OMOP Common Data Model[4] to create new column names, which include: `patient_id`, `hospital_admission_id`, `event_type` (cost_domain_id), `event_id` (cost_event_id), `chargetime`, and `cost`. For cost sampling, we take a two-step sampling procedure to simulate cost values. First, we sampled discrete-valued mean costs for four different medical event types (diagnosis, procedure, prescription, and lab events) from a Poisson distribution with a mean of 10. Then, we sampled continuous-valued costs from Gaussian distributions with their corresponding means.

**Time shifting** The time span of MIMIC-III is over a hundred years due to the de-identification process (originally seven years), while eICU's time span remains intact (two years). To simulate a more realistic time span asked in the hospital and match the time across the databases, we shifted the admission time of every patient's records ranging from 2100 to 2105. In addition, to incorporate the concept of *current* and the use of relative time expressions, we set the current time to be "2105-12-31 23:59:00" and removed any record past the current time. In this way, patients with missing hospital discharge time are considered current patients. The poll result revealed that many time expressions used in the hospital are relative time expressions (*e.g.*, today, yesterday, last month, etc.), and therefore this process was necessary to reflect time-sensitive questions. More details are reported in Supplementary D.2.

**De-identification** MIMIC-III and eICU databases are de-identified datasets, and a user needs to request credentialed access to PhysioNet[5] to obtain them. Both datasets, however, do contain real patient records, and the questions derived from them could potentially reveal patient-specific information. For example, the question "What medication was given to patient ID 1234 after being diagnosed with diabetes?" implies that this patient was diagnosed with diabetes. Combined with some unforeseen external knowledge, this might lead to recovering the patient's identity. To add another layer of de-identification, we randomly shuffled values across all patients in the database before sampling condition values. In this way, the semantic structure of the question remains the same, but sampled condition values are untraceable. More details of the de-identification process are reported in Supplementary D.3.

---

[4] http://ohdsi.github.io/CommonDataModel/cdm54.html
[5] https://physionet.org/

Table 4: The comparison between EHRSQL and other text-to-SQL datasets. *# Nesting/Q* is the average number of nesting levels per query. *% Time Used/Q* is the percentage of at least one time column used in queries. *?Schema* is whether the question authors are unaware of the database schema when they first come up with their question. *UnANS* indicates whether the dataset contains unanswerable questions.

| Dataset | # Example | # DB | # Table/DB | # Row/Table | # Table/Q | # Nesting/Q | % Time Used/Q | ?Schema | UnANS |
|---|---|---|---|---|---|---|---|---|---|
| Spider[†] | 8K | 160 | 5.1 | 2K | 1.6 | 1.2 | 12.7% | ✗ | ✗ |
| KaggleDBQA | 0.3K | 8 | 2.3 | 280K | 1.2 | 1.0 | 17.3% | ✗ | ✗ |
| SEDE | 12K | 1 | 29 | - | 1.8 | 1.3 | 20.2% | ✗ | ✗ |
| MIMICSQL | 10K | 1 | 5 | 7K | 1.8 | 1.0 | 26.6% | ✗ | ✗ |
| emrKBQA[‡] | 940K | 1 | 9 | - | - | - | - | ✓ | ✗ |
| **EHRSQL** | 24K | 2 | 13.5 | 108K | 2.4 | 2.7 | 93.2% | ✓ | ✓ |

†Train and validation sets are counted. ‡The dataset is yet to be publicly released.

## 3.3 Paraphrase Generation

To add linguistic variety to the questions, we generated template paraphrases from the question templates with manual paraphrasing and the help of machine learning models. Before paraphrasing, we ensure that the templates do not contain any domain-specific expressions, creating the paraphrasing process departing from the healthcare domain to leverage general-domain tools. The overall procedure is as follows:

1. Human paraphrasing is first conducted to add more high-quality templates, averaging 21 paraphrases per question template.

2. Based on the human paraphrases, more paraphrases are generated using machine learning models, such as T5 paraphrasers [6, 5, 1] and multilingual translation models for back-translation [8, 31].

3. The paraphrases that are too different from the original meaning are filtered using a duplicate question detection model, specifically RoBERTa-large [20] trained on the Quora duplicate question detection dataset.

4. The paraphrases are ranked by the perplexity score from GPT-Neo 1.3B [10] per question template and disregarded if they are too similar to the other paraphrases with lower perplexity. The Levenshtein distance is used to detect lexical similarity.

5. The final paraphrases are reviewed by crowd workers to rate the quality of the paraphrases. In our case, we collaborated with a crowd-sourcing company named Selectstar[6]. Three groups of annotators were assigned to this task and marked a pass or fail for each paraphrase sample. The samples with unanimous pass marks were considered the final machine-paraphrased templates, leaving 47 paraphrases per template on average.

During the paraphrasing process, slots are replaced with generic values (*e.g.*, `{patient_id}` → the patient) and returned to the original slots after the process is complete. Later, those slots are filled with the actual condition values in the value sampling process to construct the final question-to-SQL pairs. The overview of the paraphrase pipeline is illustrated in Supplementary E.

## 3.4 EHRSQL and Other Datasets

Table 4 summarizes the statistics of EHRSQL and other text-to-SQL datasets in the literature. Compared to general domain datasets (first three rows), EHRSQL has a large number of text-to-SQL pairs (*# Example*), tables per database (*# Table/DB*), and rows per table (*# Row/Table*). KaggleDBQA [19] and SEDE [12] are designed to bridge the gap between academic datasets and practical usability by using real databases and naturally-occurring utterances. However, we have gone one step further where the question authors (the poll respondents) were not presented with the database schema (*?Schema*), which adds more reality to the dataset [12]. Finally, EHRSQL contains unanswerable questions (*UnANS*) that were collected together from the poll, which may play a critical role in assessing the reliability of the QA model. To the best of our knowledge, EHRSQL is the first attempt to combine answerable and unanswerable questions in the context of text-to-SQL.

From a healthcare perspective, EHRSQL covers a variety of questions frequently asked in the hospital. The scope of these questions ranges from non-patient information (*e.g.*, the cost of a procedure) to

---

[6]https://selectstar.ai/en

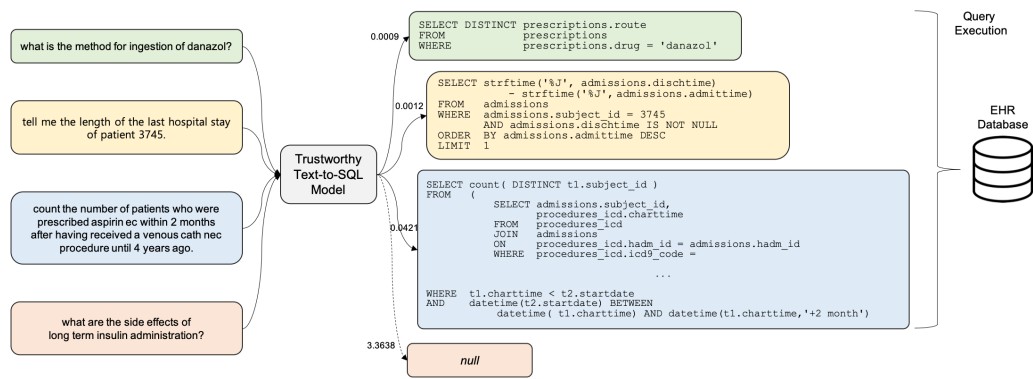

Figure 4: Overview of trustworthy semantic parsing.

individual-level information (*e.g.*, retrieving patient demographics and the lab values) and group-level information (*e.g.*, counting the number of patients in the hospital and the top N frequently prescribed drugs of patients over 60s), further combined with various time expressions (*% Time Used/Q*). The SQL queries are linked to two EHR databases: MIMIC-III and eICU, and this is the first attempt to label SQL queries on eICU, allowing questions about multi-center critical care to be answered.

# 4 Benchmarks

## 4.1 Task

In this section, we define a new text-to-SQL task that assesses models under more realistic healthcare QA settings than prior works—namely, trustworthy semantic parsing. As illustrated in Figure 4, the trustworthy model needs to 1) generate SQL queries that reflect a wide range of needs in the hospital workplace, 2) understand various time expressions to answer time-sensitive questions in healthcare, and 3) have the capacity to distinguish whether a given question is answerable or unanswerable based on the prediction confidence. Among various ways to calculate the confidence score, we argue that a desirable way to calculate it is not based on the data the model is trained on, but on the model prediction itself. As illustrated in Figure 4, once a confidence score exceeds some threshold of choice (assuming that a greater score means less confidence), the generated SQL should not be sent to the database.

To make the task feasible, we keep the condition values (*e.g.*, prescription name) the same in this work, except for genders and vital signs, as this is another major challenge in semantic parsing [30, 29]. When splitting the dataset into train, validation, and test sets, we ensure that all the question templates are present in each split. To create a more realistic QA setting, we include unanswerable questions only in the validation and test sets (consisting of 33% of each split), following the OOD detection setting [22, 39]. Among 24,411 question pairs in the dataset, the train, valid, and test splits contain 9.3K, 1.1K, and 1.8K pairs, respectively, for each database. We release the train and valid splits, totaling 21K samples, and the other 3K samples are saved for a hidden test set. Details of data splitting are reported in Supplementary F.

## 4.2 Evaluation

The goal of the trustworthy semantic parsing task is to correctly return the answers to answerable questions while disregarding unanswerable questions. This process can be evaluated in two aspects. The model first needs to have a strategy that distinguishes whether a given question is answerable or unanswerable. Given that the goal is to correctly recognize as many answerable questions as possible, the model's performance can be measured in precision and recall. In this case, the precision ($P_{ans}$) calculates the number of correctly recognized answerable questions among all questions predicted to be answerable, while recall ($R_{ans}$) calculates the number of correctly recognized answerable questions among all answerable questions. We use $F1_{ans}$ to combine these two scores.

Table 5: Performance on MIMIC-III (Left) and eICU (Right).

| Model | Valid | | | | Test | | | | Model | Valid | | | | Test | | | |
|---|---|---|---|---|---|---|---|---|---|---|---|---|---|---|---|---|---|
| | $F1_{ans}$ | $P_{exe}$ | $R_{exe}$ | $F1_{exe}$ | $F1_{ans}$ | $P_{exe}$ | $R_{exe}$ | $F1_{exe}$ | | $F1_{ans}$ | $P_{exe}$ | $R_{exe}$ | $F1_{exe}$ | $F1_{ans}$ | $P_{exe}$ | $R_{exe}$ | $F1_{exe}$ |
| | *Threshold: None* | | | | | | | | | *Threshold: None* | | | | | | | |
| T5 | 80.8 | 65.2 | 96.2 | 77.7 | 80.3 | 64.0 | 95.4 | 76.6 | T5 | 80.7 | 65.2 | 96.4 | 77.8 | 80.4 | 64.0 | 95.2 | 76.5 |
| T5 + Schema | 80.8 | 65.3 | 96.4 | 77.9 | 80.3 | 64.0 | 95.3 | 76.6 | T5 + Schema | 80.7 | 64.4 | 95.2 | 76.8 | 80.4 | 64.1 | 95.4 | 76.7 |
| | *Threshold: Clustering-based* | | | | | | | | | *Threshold: Clustering-based* | | | | | | | |
| T5 | 93.1 | 89.3 | 94.5 | 91.8 | 90.1 | 83.6 | 93.1 | 88.1 | T5 | 92.4 | 89.4 | 92.8 | 91.1 | 89.5 | 82.1 | 93.2 | 87.3 |
| T5 + Schema | 92.2 | 86.7 | 95.0 | 90.6 | 89.8 | 82.5 | 93.8 | 87.8 | T5 + Schema | 92.7 | 86.8 | 94.0 | 90.3 | 89.5 | 79.9 | 95.0 | 86.8 |
| | *Threshold: Percentile-based* | | | | | | | | | *Threshold: Percentile-based* | | | | | | | |
| T5 | 94.8 | 94.1 | 93.2 | 93.7 | 91.4 | 88.8 | 91.1 | 89.9 | T5 | 93.0 | 92.5 | 91.7 | 92.1 | 90.8 | 87.5 | 91.6 | 89.5 |
| T5 + Schema | 93.5 | 93.0 | 92.0 | 92.5 | 90.6 | 88.4 | 89.7 | 89.1 | T5 + Schema | 94.2 | 92.5 | 91.7 | 92.1 | 90.9 | 84.8 | 93.5 | 88.9 |

Next, we must evaluate how well the model generates correct SQL queries, given that some questions are considered answerable and ready to predict. The word *predict* here means the actions of generating a query and sending it to the database, where extra care is needed because the answer from the database may be directly used for clinical decision-making. Combining the concept of $F1_{ans}$ and execution accuracy in semantic parsing, $F1_{exe}$ counts only when the returned answer is correct. In other words, $F1_{exe}$ is a combined score of $P_{exe}$ and $R_{exe}$, where $P_{exe}$ is the ratio of the number of correctly answered questions to all questions predicted to be answerable and $R_{exe}$ is the ratio of the number of correctly answered questions to all answerable questions. As a result, the denominators of the precisions and recalls are the same, but the numerators are more strictly counted in metrics with $exe$. This gap between $ans$ and $exe$ reveals the model's inability to generate correct SQL queries, given that the questions are both recognized as answerable and indeed answerable.

## 4.3  Model Development

Most state-of-the-art (SOTA) semantic parsing models utilize grammar-based decoders and are only compatible with the grammars (or their subsets) defined in the Spider dataset. As the queries in EHRSQL are created in response to actual needs, most of them are incompatible with the Spider parser mainly due to time-related operators like `strftime`, `datetime` and `NULL`. Therefore, we chose general-purpose sequence-to-sequence models, T5-base [26] and T5-base with schema serialization [30, 12], as baseline models for our task. This choice aligns with a recent finding that transfer learning from pre-trained language models surpasses healthcare-specific text-to-SQL models [2]. Training details are reported in Supplementary G.

To give a model the ability to refuse to answer questions, we adopt a simple uncertainty estimation method inspired by [21, 33]. If the maximum entropy during the decoding process exceeds a pre-defined threshold, we consider this a refusal. The threshold values are determined by two heuristic approaches, and they are compared with the result obtained without refusal.

1. Clustering-based: Run K-means clustering with $k = 2$ on all maximum entropy values of validation samples, assuming that high-entropy samples are from unanswerable questions. The decision boundary between the two clusters is then used as the threshold.
2. Percentile-based: Set the threshold to the 67th percentile of the maximum entropy values of the validation samples, based on the fact that 33% of the questions in the validation set are unanswerable.

To illustrate the domain gap between general-domain and healthcare datasets, we create a subset of EHRSQL that could be parsed with the Spider parser. Then, we compare how well the SOTA cross-domain semantic parsing models generalize to healthcare text-to-SQL datasets. We choose MIMICSQL, which contains simple SQL queries that can all be parsed with the Spider grammar, and EHRSQL for the healthcare datasets. Among many SOTA models in the Spider leaderboard, we use Generation-Augmented Pre-training (GAP) [29] to test its zero-shot domain transfer performance.

## 4.4  Results and Findings

The baseline results are reported in Table 5. Among the three different threshold approaches, the percentile-based threshold performs best on both the validation and test sets. This result is an expected outcome because the ratios of the answerable and unanswerable questions in the validation and test sets are kept the same in our setting. As for the clustering-based method, an additional training or

Table 6: Performance of zero-shot cross-domain transfer with GAP in execution accuracy. Easy, medium, hard, and extra refer to Spider's SQL hardness criteria. The numbers in parenthesis indicate the number of correct cases divided by the number of answerable queries. All experiments are done on answerable questions in the validation set.

| Dataset | Easy | Medium | Hard | Extra | Skipped | All \ Skipped |
|---|---|---|---|---|---|---|
| MIMICSQL[32] | 46.7 (79/169) | 10.5 (85/809) | 0.0 (0/22) | - | - | 16.4 (164/1000) |
| EHRSQL | 23.8 (5/21) | - | 0.0 (0/23) | 0.0 (0/63) | (0/1408) | 4.7 (5/107) |
| MIMIC-III only | 16.7 (2/12) | - | 0.0 (0/15) | 0.0 (0/30) | (0/703) | 3.5 (2/57) |
| eICU only | 3.3 (3/9) | - | 0.0 (0/8) | 0.0 (0/33) | (0/705) | 6.0 (3/50) |

clustering technique is needed for the models to better discriminate the entropy values. The naive entropy values from the current models are not linearly separable across predictions (see Figure 7 in Supplementary H.4 for maximum entropy distributions). T5+Schema shows comparable performance to T5 in both databases. This result agrees with the recent finding in [12] that models trained in a single database setting do not effectively leverage schema information. Additional qualitative results are provided in Supplementary H, including SQL generation results by question complexity, time expressions, falsely executed results, and refused results.

Table 6 shows the performance of zero-shot cross-domain transfer with the GAP model. Unlike the queries in MIMICSQL, which are all parsable with the Spider grammar, EHRSQL has much more complex SQL operators and structures, with only about 7% (107/1,515) of the answerable questions in the validation set being parsable. As for model performance, GAP achieves 16.4% in the full MIMICSQL validation set and 5.8% in the subset of the validation set in EHRSQL. This result motivates again the need for a new practical text-to-SQL dataset in healthcare and QA models that can handle multiple real-world challenges in the hospital.

## 5 Conclusion and Future Direction

In this paper, we present EHRSQL, a new practical text-to-SQL dataset for question answering over structured information in EHRs. Through a poll conducted at a university hospital, we collected questions that are frequently asked on structured EHR data across various professions in the hospital. The questions reflect the actual needs in the hospital and different time expressions used in daily work, which are particularly crucial in healthcare. Additionally, we also collected unanswerable questions, which were the questions submitted by the respondents but turned out to be beyond the EHR schema or ambiguous. Finally, we manually labeled SQL queries for two open-source EHR databases—MIMIC-III and eICU—and cast these challenges as one task—trustworthy semantic parsing—where QA models should only answer the question when their predictions are confident but not otherwise.

Though we have carefully designed the dataset, there are several limitations. First, despite more than two hundred people participating in the poll, the source of the questions is from one Korean university hospital, which may not reflect every unique situation in different hospitals worldwide. Secondly, the question templates are paraphrased using general-domain paraphrasers; therefore, paraphrases with the heavy use of medical jargon could make them more realistic. Finally, SQL labels for two EHR databases might not be a sufficient number of databases to train and test a model for unseen EHR databases.

We expect numerous research directions with our dataset. The seed questions can be a valuable resource for expanding the scope of table-based healthcare QA tasks, such as creating interactive QA [36, 7] or developing it into multimodal QA datasets on EHRs. With the idea of trustworthy semantic parsing, a new class of end-to-end uncertainty-aware semantic parsing models can be proposed. As existing text-to-SQL models assume all inputs are answerable, this can be a unique venue for semantic parsing to bridge the gap between research and industrial needs.

## Acknowledgments and Disclosure of Funding

We would like to thank five anonymous reviewers for their time and insightful comments. We also thank Woochan Hwang for assisting us in clarifying our questions during the data labeling process. This work was supported by Institute of Information & communications Technology Planning & Evaluation (IITP) grant funded by the Korea government(MSIT) (No.2019-0-00075, Artificial Intelligence Graduate School Program(KAIST)), National Research Foundation of Korea (NRF) grant (NRF-2020H1D3A2A03100945) and Data Voucher grant (2021-DV-I-P-00114), funded by the Korea government (MSIT).

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
