## Supplementary Material

## A  Datasheet for Datasets

The following section is answers to questions listed in datasheets for datasets.

### A.1  Motivation

- For what purpose was the dataset created?
  EHRSQL is created to serve as a benchmark for trustworthy question answering systems on structured data in electronic health records (EHRs).

- Who created the dataset (e.g., which team, research group) and on behalf of which entity (e.g., company, institution, organization)?
  The authors of this paper.

- Who funded the creation of the dataset? If there is an associated grant, please provide the name of the grantor and the grant name and number.
  This work was supported by Institute of Information & Communications Technology Planning & Evaluation (IITP) grant (No.2019-0-00075, Artificial Intelligence Graduate School Program(KAIST)), National Research Foundation of Korea (NRF) grant (NRF-2020H1D3A2A03100945) and Data Voucher grant (2021-DV-I-P-00114), funded by the Korea government (MSIT).

### A.2  Composition

- What do the instances that comprise the dataset represent (e.g., documents, photos, people, countries)?
  EHRSQL contains natural questions and their corresponding SQL queries (text).

- How many instances are there in total (of each type, if appropriate)?
  There are about 24.4K instances (22.5K answerable; 1.9K unanswerable).

- Does the dataset contain all possible instances or is it a sample (not necessarily random) of instances from a larger set?
  We conducted a poll at a university hospital and collected a wide range of questions frequently asked on the structured EHR data. To reflect as many questions as possible, we templatized them and ensured that the final dataset contained all the question templates we created.

- What data does each instance consist of?
  The dataset contains question-SQL pairs if the question is answerable. Unanswerable questions do not have SQL labels.

- Is there a label or target associated with each instance?
  Labels are SQL queries.

- Is any information missing from individual instances? If so, please provide a description, explaining why this information is missing (e.g., because it was unavailable). This does not include intentionally removed information, but might include, e.g., redacted text.
  N/A.

- Are relationships between individual instances made explicit (e.g., users' movie ratings, social network links)?
  N/A.

- Are there recommended data splits (e.g., training, development/validation, testing)?
  See Section F.

- Are there any errors, sources of noise, or redundancies in the dataset?
  Question templates are created to have slots that are later filled with pre-defined values and records from the database. As a result, final questions can sound unnatural or be grammatically incorrect depending on the sampled values (*e.g.*, verb tense, articles, etc.).

- Is the dataset self-contained, or does it link to or otherwise rely on external resources (e.g., websites, tweets, other datasets)?

The labeled SQL queries rely on two open source databases: MIMIC-III (version 1.4)[1] and eICU (version 2.0)[2], which are accessible on PhysioNet[3].

- Does the dataset contain data that might be considered confidential (e.g., data that is protected by legal privilege or by doctor– patient confidentiality, data that includes the content of individuals' non-public communications)?
  N/A.

- Does the dataset contain data that, if viewed directly, might be offensive, insulting, threatening, or might otherwise cause anxiety?
  N/A.

- Does the dataset relate to people?
  Yes.

- Does the dataset identify any subpopulations (e.g., by age, gender)?
  EHRSQL is based on patients in MIMIC-III and eICU. MIMIC-III includes over forty thousand patients who stayed in critical care units of the Beth Israel Deaconess Medical Center between 2001 and 2012. eICU contains patients who were discharged between 2014 and 2015 in multiple critical care units in the United States.

- Is it possible to identify individuals (i.e., one or more natural persons), either directly or indirectly (i.e., in combination with other data) from the dataset?
  Even though MIMIC-III and eICU are already de-identified datasets, we further corrupted patient-specific information to avoid any chance of recovering a patient's identity. See Section D.3 for details.

- Does the dataset contain data that might be considered sensitive in any way (e.g., data that reveals race or ethnic origins, sexual orientations, religious beliefs, political opinions or union memberships, or locations; financial or health data; biometric or genetic data; forms of government identification, such as social security numbers; criminal history)?
  The dataset is already de-identified.

## A.3   Collection Process

- How was the data associated with each instance acquired?
  We collaborated with the Konyang University Hospital[4] and conducted a poll to collect real-world questions that are frequently asked on the structured EHR data.

- What mechanisms or procedures were used to collect the data (e.g., hardware apparatuses or sensors, manual human curation, software programs, software APIs)?
  We used the website SurveyMonkey[5] to create a poll and collect the responses. After the poll, we used Excel, Google Sheets, and Python to process and label the collected data.

- If the dataset is a sample from a larger set, what was the sampling strategy (e.g., deterministic, probabilistic with specific sampling probabilities)?
  When it involves sampling (*e.g.,* data splitting and patient de-identification), we sampled with a fixed random seed.

- Who was involved in the data collection process (e.g., students, crowdworkers, contractors) and how were they compensated (e.g., how much were crowdworkers paid)?
  There were three parts that required human involvement in the data collection process: poll for the question collection, SQL labeling, and quality checking for machine-paraphrased text. For the poll, we provided a $10 worth of coffee gift card to all poll respondents. For SQL labeling, the authors in the paper manually labeled SQL queries based on the database schemas. We did not hire any crowd worker for this task because the databases contain patient-specific information, and the SQL labeling process required numerous assumptions (*e.g.*, choice of SQL operations, schema linking, etc.). Lastly, we hired crowd workers to check the quality of machine paraphrased text, who were paid approximately $18 per hour.

---

[1] https://physionet.org/content/mimiciii/1.4/
[2] https://physionet.org/content/eicu-crd/2.0/
[3] https://physionet.org/
[4] https://www.kyuh.ac.kr/eng/
[5] www.surveymonkey.com

- Over what timeframe was the data collected?
  The poll was conducted in February of 2021, but the results do not depend much on the date of date collection.

- Were any ethical review processes conducted (e.g., by an institutional review board)?
  N/A.

- Does the dataset relate to people?
  Yes.

- Did you collect the data from the individuals in question directly, or obtain it via third parties or other sources (e.g., websites)?
  We directly collected the data through a poll.

- Were the individuals in question notified about the data collection?
  Yes. The poll respondents were notified about the use of data. The actual website we used for the poll is here[6]. The poll were conducted in Korean.

- Did the individuals in question consent to the collection and use of their data?
  The purpose of the poll was announced to hospital staff, and only the staff who were interested in the poll participated.

- If consent was obtained, were the consenting individuals provided with a mechanism to revoke their consent in the future or for certain uses?
  N/A.

- Has an analysis of the potential impact of the dataset and its use on data subjects (e.g., a data protection impact analysis) been conducted?
  The dataset does not have individual-specific information.

## A.4 Preprocessing/cleaning/labeling

- Was any preprocessing/cleaning/labeling of the data done (e.g., discretization or bucketing, tokenization, part-of-speech tagging, SIFT feature extraction, removal of instances, processing of missing values)?
  N/A.

- Was the "raw" data saved in addition to the preprocessed/cleaned/labeled data (e.g., to support unanticipated future uses)?
  N/A.

- Is the software that was used to preprocess/clean/label the data available?
  Preprocessing, cleaning, and labeling are done via Excel, Google Sheets, and Python.

## A.5 Uses

- Has the dataset been used for any tasks already?
  No.

- Is there a repository that links to any or all papers or systems that use the dataset?
  No.

- What (other) tasks could the dataset be used for?
  In addition to solving trustworthy semantic parsing, the seed questions themselves can be a good starting point for any healthcare table-based question answering tasks.

- Is there anything about the composition of the dataset or the way it was collected and preprocessed/cleaned/labeled that might impact future uses?
  N/A.

- Are there tasks for which the dataset should not be used?
  N/A.

---

[6] https://www.surveymonkey.com/r/Preview/?sm=hv3JkWYLdzXq2G8m_2Bh8yXI8Q_ 2FHVOzmZHcwFs7D5WhDYPQwgBHaa7OZXASgLWXsBw

## A.6 Distribution

- Will the dataset be distributed to third parties outside of the entity (e.g., company, institution, organization) on behalf of which the dataset was created?
  No.

- How will the dataset will be distributed (e.g., tarball on website, API, GitHub)?
  The dataset is released at `https://github.com/glee4810/EHRSQL`.

- When will the dataset be distributed?
  Now.

- Will the dataset be distributed under a copyright or other intellectual property (IP) license, and/or under applicable terms of use (ToU)?
  The dataset is released under MIT License.

- Have any third parties imposed IP-based or other restrictions on the data associated with the instances?
  No.

- Do any export controls or other regulatory restrictions apply to the dataset or to individual instances?
  No.

## A.7 Maintenance

- Who will be supporting/hosting/maintaining the dataset?
  The authors of this paper.

- How can the owner/curator/manager of the dataset be contacted (e.g., email address)?
  Contact the first author (`gyubok.lee@kaist.ac.kr`) or other authors.

- Is there an erratum?
  No.

- Will the dataset be updated (e.g., to correct labeling errors, add new instances, delete instances)?
  If any correction is needed, we plan to upload a new version.

- If the dataset relates to people, are there applicable limits on the retention of the data associated with the instances (e.g., were the individuals in question told that their data would be retained for a fixed period of time and then deleted)?
  N/A

- Will older versions of the dataset continue to be supported/hosted/maintained?
  We plan to maintain the newest version only.

- If others want to extend/augment/build on/contribute to the dataset, is there a mechanism for them to do so?
  Contact the authors of the paper.

# B Full List of Templates

## B.1 Question Templates

The full list of question templates is reported in Table 7. The total number of answerable question templates is 174, but a few can be unanswerable depending on the database (*e.g.*, a question about a procedure done in other hospitals is not answerable in MIMIC-III). For unanswerable questions, the template generation process explained in Section 3.1.1 is not strictly applied. As a result, unanswerable questions can contain ambiguous and too detailed questions (*e.g.*, Tell me what medicine to use to relieve a headache in hypertensive patients). We do not provide a full list of unanswerable questions as they are not subject to training and can be anything complementary to the answerable ones.

Table 7: Full list of answerable question templates.

| Patient scope | Question template | Assumption |
|---|---|---|
| None | What is the intake method of {drug_name}? | |
| None | What is the cost of a procedure named {procedure_name}? | |
| None | What is the cost of a {lab_name} lab test? | |
| None | What is the cost of a drug named {drug_name}? | |
| None | What is the cost of diagnosing {diagnosis_name}? | |
| None | What does {abbreviation} stand for? | |
| Single | What is the gender of patient {patient_id}? | |
| Single | What is the date of birth of patient {patient_id}? | |
| Single | What was the [time_filter_exact1] length of hospital stay of patient {patient_id}? | Only current patients |
| Single | What is the change in the weight of patient {patient_id} from the [time_filter_exact2] value measured [time_filter_global1]? | |
| Single | What is the change in the weight of patient {patient_id} from the [time_filter_exact2] value measured [time_filter_global2] compared to the [time_filter_exact1] value measured [time_filter_global1]? | |
| Single | What is the change in the value of {lab_name} of patient {patient_id} from the [time_filter_exact2] value measured [time_filter_global2] compared to the [time_filter_exact1] value measured [time_filter_global1]? | |
| Single | What is the change in the {vital_name} of patient {patient_id} from the [time_filter_exact2] value measured [time_filter_global2] compared to the [time_filter_exact1] value measured [time_filter_global1]? | |
| Single | Is the value of {lab_name} of patient {patient_id} [time_filter_exact2] measured [time_filter_global2] [comparison] than the [time_filter_exact1] value measured [time_filter_global1]? | |
| Single | Is the {vital_name} of patient {patient_id} [time_filter_exact2] measured [time_filter_global2] [comparison] than the [time_filter_exact1] value measured [time_filter_global1]? | |
| Single | What is_verb the age of patient {patient_id} [time_filter_global1]? | |
| Single | What is_verb the name of insurance of {patient_id} [time_filter_global1]? | |
| Single | What is_verb the marital status of patient {patient_id} [time_filter_global1]? | |
| Single | What percentile is the value of {lab_value} in a {lab_name} lab test among patients of the same age as patient {patient_id} [time_filter_global1]? | |
| Single | How many [unit_count] have passed since patient {patient_id} was admitted to the hospital currently? | Only current patient |
| Single | How many [unit_count] have passed since patient {patient_id} was admitted to the ICU currently? | Only current ICU patient |
| Single | How many [unit_count] have passed since the [time_filter_exact1] time patient {patient_id} stayed in careunit {careunit} on the current hospital visit? | Only current patient |
| Single | How many [unit_count] have passed since the [time_filter_exact1] time patient {patient_id} stayed in ward {ward_id} on the current hospital visit? | Only current patient |
| Single | How many [unit_count] have passed since the [time_filter_exact1] time patient {patient_id} received a procedure on the current hospital visit? | Only current patient |
| Single | How many [unit_count] have passed since the [time_filter_exact1] time patient {patient_id} received a {procedure_name} procedure on the current hospital visit? | Only current patient |
| Single | How many [unit_count] have passed since the [time_filter_exact1] time patient {patient_id} was diagnosed with {diagnosis_name} on the current hospital visit? | Only current patient |
| Single | How many [unit_count] have passed since the [time_filter_exact1] time patient {patient_id} was prescribed {drug_name} on the current hospital visit? | Only current patient |
| Single | How many [unit_count] have passed since the [time_filter_exact1] time patient {patient_id} received a {lab_name} lab test on the current hospital visit? | Only current patient |

Table 7: Full list of answerable question templates. (Continued)

| Patient scope | Question template | Assumption |
|---|---|---|
| Single | How many [unit_count] have passed since the [time_filter_exact1] time patient {patient_id} had a {intake_name} intake on the current ICU visit? | Only current ICU patient |
| Single | What was the [time_filter_exact1] hospital admission type of patient {patient_id} [time_filter_global1]? | |
| Single | What was the [time_filter_exact1] ward of patient {patient_id} [time_filter_global1]? | |
| Single | What was the [time_filter_exact1] careunit of patient {patient_id} [time_filter_global1]? | |
| Single | What was the [time_filter_exact1] measured height of patient {patient_id} [time_filter_global1]? | |
| Single | What was the [time_filter_exact1] measured weight of patient {patient_id} [time_filter_global1]? | |
| Single | What was the name of the diagnosis that patient {patient_id} [time_filter_exact1] received [time_filter_global1]? | |
| Single | What was the name of the procedure that patient {patient_id} [time_filter_exact1] received [time_filter_global1]? | |
| Single | What was the name of the drug that patient {patient_id} was [time_filter_exact1] prescribed via {drug_route} route [time_filter_global1]? | |
| Single | What was the name of the drug that patient {patient_id} was [time_filter_exact1] prescribed [time_filter_global1]? | |
| Single | What was the name of the drug that patient {patient_id} was prescribed [time_filter_within] after having been diagnosed with {diagnosis_name} [time_filter_global1]? | |
| Single | What was the name of the drug that patient {patient_id} was prescribed [time_filter_within] after having received a {procedure_name} procedure [time_filter_global1]? | |
| Single | What was the dose of {drug_name} that patient {patient_id} was [time_filter_exact1] prescribed [time_filter_global1]? | |
| Single | What was the total amount of dose of {drug_name} that patient {patient_id} were prescribed [time_filter_global1]? | |
| Single | What was the name of the drug that patient {patient_id} were prescribed [n_times] [time_filter_global1]? | |
| Single | What is the new prescription of patient {patient_id} [time_filter_global2] compared to the prescription [time_filter_global1]? | global filters do not overlap |
| Single | What was the [time_filter_exact1] measured value of a {lab_name} lab test of patient {patient_id}[time_filter_global1]? | |
| Single | What was the name of the lab test that patient {patient_id} [time_filter_exact1] received [time_filter_global1]? | |
| Single | what was the [agg_function] {lab_name} value of patient {patient_id} [time_filter_global1]? | |
| Single | What was the name of the allergy that patient {patient_id} had [time_filter_global1]? | |
| Single | What was the name of the substance that patient {patient_id} was allergic to [time_filter_global1]? | |
| Single | What was the organism name found in the [time_filter_exact1] {culture_name} microbiology test of patient {patient_id} [time_filter_global1]? | |
| Single | What was the name of the specimen that patient {patient_id} was [time_filter_exact1] tested [time_filter_global1]? | |
| Single | What was the name of the intake that patient {patient_id} [time_filter_exact1] had [time_filter_global1]? | |
| Single | What was the total volume of {intake_name} intake that patient {patient_id} received [time_filter_global1]? | |
| Single | What was the total volume of intake that patient {patient_id} received [time_filter_global1]? | |

Table 7: Full list of answerable question templates. (Continued)

| Patient scope | Question template | Assumption |
|---|---|---|
| Single | What was the name of the output that patient {patient_id} [time_filter_exact1] had [time_filter_global1]? | |
| Single | What was the total volume of {output_name} output that patient {patient_id} had [time_filter_global1]? | |
| Single | What was the total volume of output that patient {patient_id} had [time_filter_global1]? | |
| Single | What is the difference between the total volume of intake and output of patient {patient_id} [time_filter_global1]? | |
| Single | What was the [time_filter_exact1] measured {vital_name} of patient {patient_id}[time_filter_global1]? | |
| Single | What was the {agg_function} {vital_name} of patient {patient_id}[time_filter_global1]? | |
| Single | What is_verb the total hospital cost of patient {patient_id} [time_filter_global1]? | |
| Single | When was the [time_filter_extract1] hospital admission time of patient {patient_id} [time_filter_global1]? | |
| Single | When was the [time_filter_extract1] hospital admission time that patient {patient_id} was admitted via {admission_route}[time_filter_global1]? | |
| Single | When was the [time_filter_extract1] hospital discharge time of patient {patient_id}[time_filter_global1]? | |
| Single | When was the [time_filter_extract1] length of ICU stay of patient {patient_id}? | No current ICU patient |
| Single | When was the [time_filter_extract1] time that patient {patient_id} was diagnosed with {diagnosis_name} [time_filter_global1]? | |
| Single | When was the [time_filter_extract1] procedure time of patient {patient_id} [time_filter_global1]? | |
| Single | When was the [time_filter_extract1] time that patient {patient_id} received a {procedure_name} procedure [time_filter_global1]? | |
| Single | When was the [time_filter_extract1] prescription time of patient {patient_id} [time_filter_global1]? | |
| Single | When was the [time_filter_extract1] time that patient {patient_id} was prescribed {drug_name} [time_filter_global1]? | |
| Single | When was the [time_filter_extract1] time that patient {patient_id} was prescribed {drug_name1} and {drug_name2} [time_filter_within] [time_filter_global1]? | |
| Single | When was the [time_filter_extract1] time that patient {patient_id} was prescribed a medication via {drug_route} route [time_filter_global1]? | |
| Single | When was the [time_filter_extract1] time that patient {patient_id} was prescribed a medication via {drug_route} route [time_filter_global1]? | |
| Single | When was the [time_filter_extract1] lab test of patient {patient_id} [time_filter_global1]? | |
| Single | When was the [time_filter_extract1] time that patient {patient_id} received a {lab_test} lab test [time_filter_global1]? | |
| Single | When was the [time_filter_extract1] time that patient {patient_id} had the [sort] value of {lab_name} [time_filter_global1]? | |
| Single | When was the [time_filter_extract1] microbiology test of patient {patient_id} [time_filter_global1]? | |
| Single | When was patient {patient_id}'s [time_filter_extract1] {culture_name} microbiology test [time_filter_global1]? | |
| Single | When was the [time_filter_extract1] time that patient {patient_id} had a {intake_name} intake [time_filter_global1]? | |
| Single | When was the [time_filter_extract1] intake time of patient {patient_id} [time_filter_global1]? | |

Table 7: Full list of answerable question templates. (Continued)

| Patient scope | Question template | Assumption |
|---|---|---|
| Single | When was the [time_filter_extract1] time that patient {patient_id} had a {output_name} output [time_filter_global1]? | |
| Single | When was the [time_filter_extract1] time that patient {patient_id} had a {vital_name} measured [time_filter_global1]? | |
| Single | When was the [time_filter_extract1] time that the {vital_name} of patient {patient_id} was [comparison] than {vital_value} [time_filter_global1]? | |
| Single | When was the [time_filter_extract1] time that patient {patient_id} had the [sort] {vital_name} [time_filter_global1]? | |
| Single | Has_verb patient {patient_id} received a {procedure_name} procedure in other than the current hospital [time_filter_global1]? | Only sample current patient |
| Single | Has_verb patient {patient_id} been admitted to the hospital [time_filter_global1]? | |
| Single | Has_verb patient {patient_id} been to an emergency room [time_filter_global1]? | |
| Single | Has_verb patient {patient_id} received any procedure [time_filter_global1]? | |
| Single | Has_verb patient {patient_id} received a {procedure_name} procedure [time_filter_global1]? | |
| Single | What was the name of the procedure that patient {patient_id} received [n_times] [time_filter_global1]? | |
| Single | Has_verb patient {patient_id} received any diagnosis [time_filter_global1]? | |
| Single | Has_verb patient {patient_id} been diagnosed with {diagnosis_name} [time_filter_global1]? | |
| Single | Has_verb patient {patient_id} been prescribed {drug_name1},{drug_name2}, or {drug_name3} [time_filter_global1]? | |
| Single | Has_verb patient {patient_id} been prescribed any medication [time_filter_global1]? | |
| Single | Has_verb patient {patient_id} been prescribed {drug_name} [time_filter_global1]? | |
| Single | Has_verb patient {patient_id} received any lab test [time_filter_global1]? | |
| Single | Has_verb patient {patient_id} received a {lab_name} lab test [time_filter_global1]? | |
| Single | Has_verb patient {patient_id} had any allergy [time_filter_global1]? | |
| Single | Has_verb patient {patient_id} had any microbiology test result [time_filter_global1]? | |
| Single | Has_verb patient {patient_id} had any {culture_name} microbiology test result [time_filter_global1]? | |
| Single | Has_verb there been any organism found in the [time_filter_extract1]{culture_name} microbiology test of patient {patient_id} [time_filter_global1]? | |
| Single | Has_verb patient {patient_id} had any {intake_name} intake [time_filter_global1]? | |
| Single | Has_verb patient {patient_id} had any {output_name} output [time_filter_global1]? | |
| Single | Has_verb the {vital_name} of patient {patient_id} been ever [comparison] than {vital_value} [time_filter_global1]? | |
| Single | Has_verb the {vital_name} of patient {patient_id} been normal [time_filter_global1]? | |
| Single | List the hospital admission time of patient {patient_id} [time_filter_global1]. | |
| Single | List the [unit_average] [agg_function] {lab_name} lab value of patient {patient_id} [time_filter_global1]. | |
| Single | List the [unit_average] [agg_function] weight of patient {patient_id} [time_filter_global1]. | |
| Single | List the [unit_average] [agg_function] volume of {intake_name} intake that patient {patient_id} received [time_filter_global1]. | |
| Single | List the [unit_average] [agg_function] volume of {output_name} output that patient {patient_id} had [time_filter_global1]. | |

Continued on next page

Table 7: Full list of answerable question templates. (Continued)

| Patient scope | Question template | Assumption |
|---|---|---|
| Single | List the [unit_average] [agg_function] {vital_name} of patient {patient_id} [time_filter_global1]. | |
| Single | Count the number of hospital visits of patient {patient_id} [time_filter_global1]. | |
| Single | Count the number of ICU visits of patient {patient_id} [time_filter_global1]. | |
| Single | Count the number of times that patient {patient_id} received a {procedure_name} procedure [time_filter_global1]. | |
| Single | Count the number of drugs patient {patient_id} was prescribed [time_filter_global1]. | |
| Single | Count the number of times that patient {patient_id} were prescribed {drug_name} [time_filter_global1]. | |
| Single | Count the number of times that patient {patient_id} received a {lab_name} lab test [time_filter_global1]. | |
| Single | Count the number of times that patient {patient_id} had a {intake_name} intake [time_filter_global1]. | |
| Single | Count the number of times that patient {patient_id} had a {output_name} output [time_filter_global1]. | |
| Group | Count the number of current patients. | |
| Group | Count the number of current patients aged [age_group]. | |
| Group | What is the [n_survival_period] survival rate of patients diagnosed with {diagnosis_name}? | |
| Group | What is the [n_survival_period] survival rate of patients who were prescribed {drug_name} after having been diagnosed with {diagnosis_name}? | |
| Group | What are the top [n_rank] diagnoses that have the highest [n_survival_period] mortality rate? | |
| Group | What is_verb the [agg_fuction] total hospital cost that involves a procedure named {procedure_name} [time_filter_global1]? | |
| Group | What is_verb the [agg_fuction] total hospital cost that involves a {lab_name} lab test [time_filter_global1]? | |
| Group | What is_verb the [agg_fuction] total hospital cost that involves a drug named {drug_name} [time_filter_global1]? | |
| Group | What is_verb the [agg_fuction] total hospital cost that involves a diagnosis named {diagnosis_name} [time_filter_global1]? | |
| Group | List the IDs of patients diagnosed with {diagnosis_name} [time_filter_global1]. | |
| Group | What is_verb the [agg_fuction] [unit_average] number of patient records diagnosed with {diagnosis_name} [time_filter_global1]? | |
| Group | Count the number of patients who were dead after having been diagnosed with {diagnosis_name} [time_filter_within] [time_filter_global1]. | |
| Group | Count the number of patients who did not come back to the hospital [time_filter_within] after diagnosed with {diagnosis_name} [time_filter_global1]. | |
| Group | Count the number of patients who were admitted to the hospital [time_filter_global1]. | |
| Group | Count the number of patients who were discharged from the hospital [time_filter_global1]. | |
| Group | Count the number of patients who stayed in ward {ward_id} [time_filter_global1]. | |
| Group | Count the number of patients who stayed in careunit{careunit} [time_filter_global1]. | |
| Group | Count the number of patients who were diagnosed with{diagnosis_name} [time_filter_within] after having received a {procedure_name} procedure [time_filter_global1]. | |
| Group | Count the number of patients who were diagnosed with{diagnosis_name2} [time_filter_within] after having been diagnosed with {diagnosed_name1} [time_filter_global1]. | |
| Group | Count the number of patients who were diagnosed with {diagnosis_name} [time_filter_global1]. | |
| Group | Count the number of patients who received a {procedure_name} procedure [time_filter_global1]. | |

Continued on next page

Table 7: Full list of answerable question templates. (Continued)

| Patient scope | Question template | Assumption |
|---|---|---|
| Group | Count the number of patients who received a {procedure_name} procedure [n_times] [time_filter_global1]. | |
| Group | Count the number of patients who received a {procedure_name2} procedure [time_filter_within] after having received a {procedure_name1} procedure [time_filter_global1]. | |
| Group | Count the number of patients who received a {procedure_name} procedure [time_filter_within] after having been diagnosed with {diagnosis_name1} [time_filter_global1]. | |
| Group | Count the number of {procedure_name} procedure cases [time_filter_global1]. | |
| Group | Count the number of patients who were prescribed {drug_name} [time_filter_global1]. | |
| Group | Count the number of {drug_name} prescription cases [time_filter_global1]. | |
| Group | Count the number of patients who were prescribed {drug_name} [time_filter_within] after having received a {procedure_name} procedure [time_filter_global1]. | |
| Group | Count the number of patients who were prescribed {drug_name} [time_filter_within] after having been diagnosed with {diagnosis_name} [time_filter_global1]. | |
| Group | Count the number of patients who received a {lab_name} lab test [time_filter_global1]. | |
| Group | Count the number of patients who received a {culture_name} microbiology test [time_filter_global1]. | |
| Group | Count the number of patients who had a {intake_name} intake [time_filter_global1]. | |
| Group | What are_verb the top [n_rank] frequent diagnoses [time_filter_global1]? | |
| Group | What are_verb the top [n_rank] frequent diagnoses of patients aged [age_group] [time_filter_global1]? | |
| Group | What are_verb the top [n_rank] frequent diagnoses that patients were diagnosed [time_filter_within] after having received a {procedure_name} procedure [time_filter_global1]? | |
| Group | What are_verb the top [n_rank] frequent diagnoses that patients were diagnosed [time_filter_within] after having been diagnosed with {diagnosis_name} [time_filter_global1]? | |
| Group | What are_verb the top [n_rank] frequent procedures [time_filter_global1]? | |
| Group | What are_verb the top [n_rank] frequent procedures of patients aged [age_group] [time_filter_global1]? | |
| Group | What are_verb the top [n_rank] frequent procedures that patients received [time_filter_within] after having received a {procedure_name} procedure [time_filter_global1]? | |
| Group | What are_verb the top [n_rank] frequent procedures that patients received [time_filter_within] after having been diagnosed with {diagnosis_name} [time_filter_global1]? | |
| Group | What are_verb the top [n_rank] frequently prescribed drugs [time_filter_global1]? | |
| Group | What are_verb the top [n_rank] frequently prescribed drugs of patients aged [age_group] [time_filter_global1]? | |
| Group | What are_verb the top [n_rank] frequent prescribed drugs for patients who were also prescribed {drug_name} [time_filter_within] [time_filter_global1]? | |
| Group | What are_verb the top [n_rank] frequent drugs that patients were prescribed [time_filter_within] after having been prescribed with {drug_name} [time_filter_global1]? | |
| Group | What are_verb the top [n_rank] frequent drugs that patients were prescribed [time_filter_within] after having received a {procedure_name} procedure [time_filter_global1]? | |
| Group | What are_verb the top [n_rank] frequent drugs that patients were prescribed [time_filter_within] after having been diagnosed with {diagnosis_name} [time_filter_global1]? | |
| Group | What are_verb the top [n_rank] frequently prescribed drugs that patients aged [age_group] were prescribed [time_filter_within] after having been diagnosed with {diagnosis_name} [time_filter_global1]? | |
| Group | What are_verb the top [n_rank] frequently prescribed drugs that {gender} patients aged [age_group] were prescribed [time_filter_within] after having been diagnosed with {diagnosis_name} [time_filter_global1]? | |

Table 7: Full list of answerable question templates. (Continued)

| Patient scope | Question template | Assumption |
|---|---|---|
| Group | What are_verb the top [n_rank] frequent lab test [time_filter_global1]? | |
| Group | What are_verb the top [n_rank] frequent lab tests of patients aged [age_group] [time_filter_global1]? | |
| Group | What are_verb the top [n_rank] frequent lab tests that patients had [time_filter_within] after having been diagnosed with {diagnosis_name} [time_filter_global1]? | |
| Group | What are_verb the top [n_rank] frequent lab tests that patients had [time_filter_within] after having received a {procedure_name} procedure [time_filter_global1]? | |
| Group | What are_verb the top [n_rank] frequent specimens tested [time_filter_global1]? | |
| Group | What are_verb the top [n_rank] frequent specimens that patients were tested [time_filter_within] after having been diagnosed with {diagnosis_name} [time_filter_global1]? | |
| Group | What are_verb the top [n_rank] frequent specimens that patients were tested [time_filter_within] after having received a {procedure_name} procedure [time_filter_global1]? | |
| Group | What are_verb the top [n_rank] frequent intake events [time_filter_global1]? | |
| Group | What are_verb the top [n_rank] frequent output events [time_filter_global1]? | |

Several question templates assume a specific range of patients (*e.g.*, current patients or already discharged patients), as indicated in the "Assumption" column in Table 7. For example, depending on the patients, the way to calculate the duration of hospital stay can be different. The SQL query for already discharged patients (*i.e.*, What was the [time_filter_exact1] length of hospital stay of patient patient_id?) calculates the time between the hospital admission and discharge, while the same query for the current patients (*i.e.*, How many [unit_count] have passed since patient patient_id was admitted to the hospital currently?) calculates the time between the hospital admission and current time. We intentionally separated the templates to make the model better understand the hidden assumptions behind user utterances.

Time slots with numbering (*e.g.*, [time_filter_global1] and [time_filter_global2]) are the variants of the time slot without numbering (*e.g.*, [time_filter_global]). The purpose of the numbering is to indicate the temporal order of time filters. Specifically, a higher number indicates the same or a later time for [time_filter_global1] and [time_filter_global2], and [time_filter_exact2] must be later than [time_filter_exact1].

Some verbs that end with "_verb" in question templates indicate that the verb tense can change depending on the sampled time templates.

### B.2 Time Templates

Table 8 shows the full list of time templates with natural language (NL) time expressions and SQL time patterns. Based on the time filter types present in question templates, time templates are sampled, and their corresponding NL time expressions and SQL time patterns are added to the question templates and SQL queries. Column slots in the SQL time patterns such as [time_column] and [hospital_dischargetime] are replaced with the actual column names following the database schema. The blanks for the NL time expressions and SQL time patterns in the table indicate that no time filter is applied.

Table 8: Full list of NL time expressions and their corresponding SQL time patterns.

| Time filter type | Expression type | Unit | Interval type | Option | NL time expression | SQL time pattern |
|---|---|---|---|---|---|---|
| global | - | - | - | - | | |
| global | relative | hospital | in | first | on the first hospital visit | WHERE [hospital_dischargetime] IS NOT NULL ORDER BY [hospital_admittime] ASC LIMIT 1 |
| global | relative | hospital | in | last | on the last hospital visit | WHERE [hospital_dischargetime] IS NOT NULL ORDER BY [hospital_admittime] DESC LIMIT 1 |

Table 8: Full list of NL time expressions and their corresponding SQL time patterns. (Continued)

| Time filter type | Expression type | Unit | Interval type | Option | NL time expression | SQL time pattern |
|---|---|---|---|---|---|---|
| global | relative | hospital | in | current | on the current hospital visit | WHERE [hospital_dischargetime] IS NULL |
| global | relative | ICU | in | first | on the first ICU visit | WHERE [icu_dischargetime] IS NOT NULL ORDER BY [icu_admittime] ASC LIMIT 1 |
| global | relative | ICU | in | last | on the last ICU visit | WHERE [icu_dischargetime] IS NOT NULL ORDER BY [icu_admittime] DESC LIMIT 1 |
| global | relative | ICU | in | current | on the current ICU visit | WHERE [icu_dischargetime] IS NULL |
| global | relative | year | in | last | last year | WHERE datetime([time_column],'start of year') = datetime(current_time,'start of year ','-1 year') |
| global | relative | year | until | last | until last year | WHERE datetime([time_column],'start of year') <= datetime(current_time,'start of year ','-1 year') |
| global | relative | year | since | last | since last year | WHERE datetime([time_column],'start of year') >= datetime(current_time,'start of year ','-1 year') |
| global | relative | year | in | this | this year | WHERE datetime([time_column],'start of year') = datetime(current_time,'start of year ','-0 year') |
| global | relative | year | until | - | until {year} year ago | WHERE datetime([time_column]) <= datetime(current_time,'-{year} year') |
| global | relative | year | since | - | since {year} year ago | WHERE datetime([time_column]) >= datetime(current_time,'-{year} year') |
| global | relative | month | in | last | last month | WHERE datetime([time_column],'start of month') = datetime(current_time,'start of month','-1 month') |
| global | relative | month | until | last | until last month | WHERE datetime([time_column],'start of month') <= datetime(current_time,'start of month','-1 month') |
| global | relative | month | since | last | since last month | WHERE datetime([time_column],'start of month') >= datetime(current_time,'start of month','-1 month') |
| global | relative | month | in | this | this month | WHERE datetime([time_column],'start of month') = datetime(current_time,'start of month','-0 month') |
| global | relative | month | until | - | until {month} month ago | WHERE datetime([time_column]) <= datetime(current_time,'-{month} month') |
| global | relative | month | since | - | since {month} month ago | WHERE datetime([time_column]) >= datetime(current_time,'-{month} month') |
| global | relative | day | in | last | yesterday | WHERE datetime([time_column],'start of day') = datetime(current_time,'start of day','-1 day') |
| global | relative | day | in | last | until yesterday | WHERE datetime([time_column],'start of day') <= datetime(current_time,'start of day','-1 day') |
| global | relative | day | in | last | since yesterday | WHERE datetime([time_column],'start of day') >= datetime(current_time,'start of day','-1 day') |
| global | relative | day | in | this | today | WHERE datetime([time_column],'start of day') = datetime(current_time,'start of day','-0 day') |
| global | relative | day | until | - | until {day} day ago | WHERE datetime([time_column]) <= datetime(current_time,'-{day} day') |
| global | relative | day | since | - | since {day} day ago | WHERE datetime([time_column]) >= datetime(current_time,'-{day} day') |
| global | absolute | year | in | - | in {year} | WHERE strftime('%Y',[time_column])= '{year}' |
| global | absolute | year | until | - | until {year} | WHERE strftime('%Y',[time_column]) <= '{year}' |
| global | absolute | year | since | - | since {year} | WHERE strftime('%Y',[time_column]) >= '{year}' |
| global | absolute | month | in | - | in {month}/{year} | WHERE strftime('%Y-%m',[time_column])= '{year}-{month}' |

Table 8: Full list of NL time expressions and their corresponding SQL time patterns. (Continued)

| Time filter type | Expression type | Unit | Interval type | Option | NL time expression | SQL time pattern |
|---|---|---|---|---|---|---|
| global | absolute | month | until | - | until {month}/{year} | `WHERE strftime('%Y-%m',[time_column]) <= '{year}-{month}'` |
| global | absolute | month | since | - | since {month}/{year} | `WHERE strftime('%Y-%m',[time_column]) >= '{year}-{month}'` |
| global | absolute | day | in | - | on {month}/{day}/{year} | `WHERE strftime('%Y-%m-%d',[time_column]) = '{year}-{month}-{day}'` |
| global | absolute | day | until | - | until {month}/{day}/{year} | `WHERE strftime('%Y-%m-%d',[time_column]) <= '{year}-{month}-{day}'` |
| global | absolute | day | since | - | since {month}/{day}/{year} | `WHERE strftime('%Y-%m-%d',[time_column]) >= '{year}-{month}{day}'` |
| global | mix | month | in | last | in {month}/last year | `WHERE datetime([time_column],'start of year') = datetime(current_time,'start of year','-1 year') AND strftime('%m',[time_column]) = '{month}'` |
| global | mix | month | in | this | in {month}/this year | `WHERE datetime([time_column],'start of year') = datetime(current_time,'start of year','-0 year') AND strftime('%m',[time_column]) = '{month}'` |
| global | mix | day | in | last | on {month}/{day}/last year | `WHERE datetime([time_column],'start of year') = datetime(current_time,'start of year','-1 year') AND strftime('%m-%d',[time_column]) = '{month}-{day}'` |
| global | mix | day | in | this | on {month}/{day}/this year | `WHERE datetime([time_column],'start of year') = datetime(current_time,'start of year','-0 year') AND strftime('%m-%d',[time_column]) = '{month}-{day}'` |
| global | mix | day | in | last | on last month/{day} | `WHERE datetime([time_column],'start of month') = datetime(current_time,'start of month','-1 month') AND strftime('%d',[time_column]) = '{day}'` |
| global | mix | day | in | this | on this month/{day} | `WHERE datetime([time_column],'start of month') = datetime(current_time,'start of month','-0 month') AND strftime('%d',[time_column]) = '{day}'` |
| within | - | - | - | - | | |
| within | - | hospital | in | - | within the same hospital visit | `WHERE [hospital_admission_id1] = [hospital_admission_id2]` |
| within | - | ICU | in | - | within the same icu visit | `WHERE [icu_admission_id1] = [icu_admission_id2]` |
| within | - | year | in | - | within the same year | `WHERE datetime([time_column1],'start of year') = datetime([time_column2],'start of year'` |
| within | - | n_year | in | - | within {year} year | `WHERE datetime([time_column2]) BETWEEN datetime([time_column1] AND datetime([time_column1],'+{year} year'` |
| within | - | month | in | - | within the same month | `WHERE datetime([time_column1],'start of month') = datetime([time_column2],'start of month'` |
| within | - | n_month | in | - | within {month} month | `WHERE datetime([time_column2]) BETWEEN datetime([time_column1] AND datetime([time_column1],'+{month} month'` |
| within | - | day | in | - | within the same day | `WHERE datetime([time_column1],'start of day') = datetime([time_column2],'start of day'` |
| within | - | n_day | in | - | within {day} day | `WHERE datetime([time_column2]) BETWEEN datetime([time_column1] AND datetime([time_column1],'+{day} day'` |
| within | - | exact | in | - | at the same time | `WHERE datetime([time_column1]) = datetime([time_column2])` |
| exact | relative | exact | at | - | first | `ORDER BY [time_column] ASC LIMIT 1` |
| exact | relative | exact | at | - | second | `ORDER BY [time_column] ASC LIMIT 1 OFFSET 1` |
| exact | relative | exact | at | - | second to last | `ORDER BY [time_column] DESC LIMIT 1 OFFSET 1` |

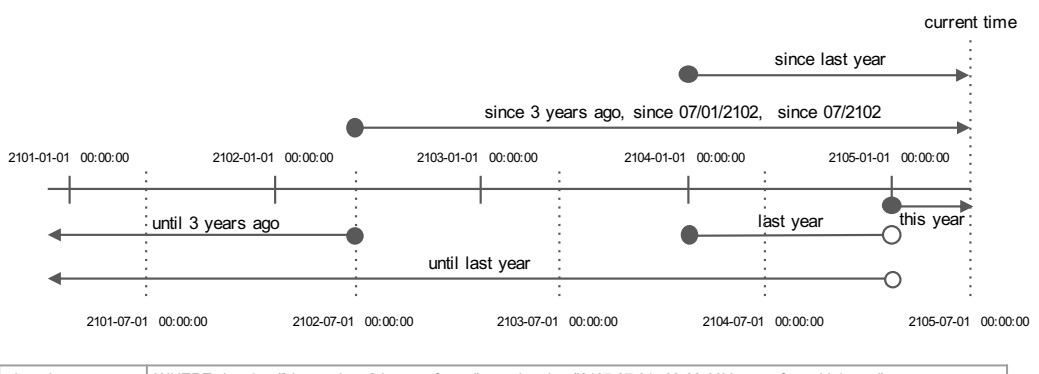

| | |
|---|---|
| since last year | WHERE datetime([time_column],'start of year') >= datetime('2105-07-01 00:00:00','start of year','-1 year') |
| since 3 years ago | WHERE datetime([time_column]) >= datetime('2105-07-01 00:00:00','-3 year') |
| since 07/01/2102 | WHERE strftime('%Y-%m-%d',[time_column]) >= '2105-07-01' |
| since 07/2102 | WHERE strftime('%Y-%m',[time_column]) >= '2105-07' |
| this year | WHERE datetime([time_column],'start of year') = datetime('2105-07-01 00:00:00','start of year','-0 year') |
| last year | WHERE datetime([time_column],'start of year') = datetime('2105-07-01 00:00:00','start of year','-1 year') |
| until 3 years ago | WHERE datetime([time_column]) <= datetime('2105-07-01 00:00:00','-3 year') |
| until last year | WHERE datetime([time_column],'start of year') <= datetime('2105-07-01 00:00:00','start of year','-1 year') |

Figure 4: Illustration of time templates.

Table 8: Full list of NL time expressions and their corresponding SQL time patterns. (Continued)

| Time filter type | Expression type | Unit | Interval type | Option | NL time expression | SQL time pattern |
|---|---|---|---|---|---|---|
| exact | relative | exact | at | - | last | ORDER BY [time_column1] DESC LIMIT 1 |
| exact | absolute | exact | at | - | at {year}-{month}-{day} {hour} :{minute}:{second} | WHERE datetime([time_column]) = '{year}-{month}-{day} {hour}:{minute} :{second}' |

Time templates with the option "this" are not combined with the interval type of "since" or "until" because combining "this" with "since" is equivalent to combining "this" with "in" (*i.e.*, since this year is equivalent to this year). Additionally, combining "this" with "until" is equivalent to no time constraint (*i.e.*, until this year is equivalent to no time filter).

In relative time expressions, the concept of N units before the current time is ambiguous because one year ago and the last year can differ depending on the context. Therefore, we strictly define "N units ago" as a time point exactly N units before the current time (*e.g.*, one year ago of the current time 2105-12-31 23:59:00 is 2104-12-31 23:59:00) and they can be combined with the "until" and "since" interval types. Figure 4 illustrates several time templates.

## B.3 Template Combination

As question templates can be combined with multiple slots, we tagged each question to track what time templates and pre-defined values are combined to form the final question. Each tag (*Q_tag*, *O_tag*, and *T_tag*) represents Stages 0, 1, and 2 in Figure 3. Except for *Q_tag*, which indicates the question template, *O_tag* and *T_tag* have fixed numbers of placeholders that store each sampled template and value. *O_tag* stores nine different types of operation values in a tuple: ([age_group], [agg_function], [comparison], [n_rank], [n_survival_period], [n_times], [sort], [unit_average], [unit_count]), sampled in Stage 1. *T_tag* stores time templates in a tuple: ([time_filter_global1], [time_filter_global2], [time_filter_within], [time_filter_exact1], [time_filter_exact2]), sampled in Stage 2.

A full list of the operation values is reported in Table 9. Similar to time templates, the operation values have both NL expressions and SQL patterns.

Table 9: Pre-defined operation values.

| Operation value type | NL operation expression | SQL operation pattern |
|---|---|---|
| [age_group] | 20's | WHERE [age_col] BETWEEN 20 AND 29 |
| | 30's | WHERE [age_col] BETWEEN 30 AND 39 |
| | 40's | WHERE [age_col] BETWEEN 40 AND 49 |
| | 50's | WHERE [age_col] BETWEEN 50 AND 59 |
| | 60 or above | WHERE [age_col] >= 60 |
| [age_function] | maximum | MAX |
| | minimum | MIN |
| | average | AVG |
| [comparison] | greater | > |
| | less | < |
| [n_rank] | three | 3 |
| | four | 4 |
| | five | 5 |
| [n_survival_period] | one year | 1 * 365 |
| | two year | 2 * 365 |
| | three year | 3 * 365 |
| | four year | 4 * 365 |
| | five year | 5 * 365 |
| [n_times] | two times | = 2 |
| | two or more times | >= 2 |
| [sort] | min | ORDER BY [sort_col] ASC LIMIT 1 |
| | max | ORDER BY [sort_col] DESC LIMIT 1 |
| [unit_average] | yearly | GROUP BY strftime('%Y',[time_col]) |
| | monthly | GROUP BY strftime('%Y-%m',[time_col]) |
| | daily | GROUP BY strftime('%Y-%m-%d',[time_col]) |
| [unit_count] | days | 1 * |
| | hours | 24 * |

## C  SQL Labeling Details

Since the questions were collected independently of the database schema, our SQL labeling process required us to make numerous assumptions. Below is a list of the assumptions we made to label the queries.

### C.1  Shared Assumptions

- The age of a patient is calculated only once at each hospital admission time. Therefore, even if a patient stays more than a year without hospital discharge, the age remains the same.

- To count the number of patients or hospital (or ICU) visits, DISTINCT is used in the SELECT clause.

- The queries about the cost of or drug routes use DISTINCT.

- When retrieving a lab value or vital sign, only the value is returned, not the unit of measurement.

- DENSE_RANK is used for ranking questions, meaning multiple items with the same ranks can be returned together. For example, a query asking about the top three frequent diagnoses may retrieve more than three diagnosis names when items with the same rank exist in the answer. Additionally, the retrieved results can be fewer than the expected number N when the number of diagnoses under some conditions is smaller than the expected number.

- When a question is related to both death and diagnosis, only the first diagnosis time is considered.

- When calculating the N-year survival rate, if a death record exists between the first diagnosis time and N years later, it counts as death. But if there is no death record within N years or the death happens after N years, it counts as survived.

- The current time and normal ranges of vital signs are post-processed after SQL generation so that they are independent of the modeling pipeline when the value changes.

- The vital signs we consider in the dataset are body temperature, SaO2, heart rate, respiratory rate, and blood pressures (systemic systolic, diastolic, and mean).

### C.1.1 Assumptions in MIMIC-III

- Diagnosis and procedure times are not available in the original database. To address this, we manually set diagnosis time as hospital admission time and procedure time as hospital discharge time. Thus, questions asking about current patients' procedure time are excluded in the MIMIC-III questions.
- Among many items in the CHARTEVENTS table, we only use weight, height, and the seven vital sign values.
- We use INPUTEVENTS_CV instead of INPUTEVENTS_MV for input events as it contains more records and one time column per record (charttime), which is more closely aligned with eICU's intakeoutput table.
- For input and output events, we only use values stored in milliliters (mL) in the case of numerical reasoning within or between input and output events.

### C.1.2 Assumptions in eICU

- Diagnosis and treatment tables have path-based names for each record. Instead of using them directly, the paths are further pre-processed to have shorter names.
- For vital signs, we choose the vitalperiodic table as it contains more records.
- Questions about drug dose are not considered in eICU as drug doses are stored in free-text (values and units are mixed).

### C.2 Mapping Between Condition Value Slots and Column Names

The mapping between condition value slots and column names in both MIMIC-III and eICU is shown in Table 10.

Table 10: Schema mapping in both MIMIC-III and eICU.

| Condition value slots | MIMIC-III | eICU |
|---|---|---|
| {abbreviation} | d_icd_procedures.short_title d_icd_diagnoses.short_title | - |
| {admission_route} | admissions.admission_location | patient.hospitaladmitsource |
| {careunit} | transfers.curr_careunit | - |
| {culture_name} | microbiologyevents.spec_type_desc | microlab.culturesite |
| {diagnosis_name} | d_icd_diagnoses.short_title | diagnosis.diagnosisname |
| {drug_name} | prescriptions.drug_name | medication.drugname |
| {drug_route} | prescriptions.route | medication.routeadmin |
| {gender} | patients.gender | patient.gender |
| {intake_name} | d_items.label | intakeoutput.celllabel |
| {lab_name} | d_labitems.label | lab.labname |
| {lab_value} | labevents.valuenum | lab.labresult |
| {output_name} | d_items.label | intakeoutput.celllabel |
| {patient_id} | patients.subject_id | patient.uniquepid |
| {procedure_name} | d_icd_procedures.short_title | treatment.treatmentname |
| {vital_name} | d_items.label | - |
| {vital_value} | chartevents.valuenum | vitalperiodic.temperature, vitalperiodic.sao2, vitalperiodic.heartrate, vitalperiodic.respiration, vitalperiodic.systemicsystolic, vitalperiodic.systemicdiastolic, vitalperiodic.systemicmean |
| {ward_id} | transfers.curr_wardid | patient.wardid |

<table>
<tr><td><Question></td><td>How much of a difference is there in patient 2518's weight last measured on the current hospital visit compared to the second to last value measured on the current hospital visit?</td></tr>
</table>

### <JOIN-based query>

```
SELECT
(
    SELECT chartevents.valuenum
    FROM chartevents
    JOIN d_items
    ON chartevents.itemid = d_items.itemid
    JOIN icustays
    ON chartevents.icustay_id = icustays.icustay_id
    JOIN admissions
    ON icustays.hadm_id = admissions.hadm_id
    WHERE admissions.subject_id = 2518
    AND d_items.label = 'admit wt'
    AND d_items.linksto = 'chartevents'
    AND admissions.dischtime IS NULL
    ORDER BY chartevents.charttime DESC LIMIT 1
)
-
(
    SELECT chartevents.valuenum
    FROM chartevents
    JOIN d_items
    ON chartevents.itemid = d_items.itemid
    JOIN icustays
    ON chartevents.icustay_id = icustays.icustay_id
    JOIN admissions
    ON icustays.hadm_id = admissions.hadm_id
    WHERE admissions.subject_id = 2518
    AND d_items.label = 'admit wt'
    AND d_items.linksto = 'chartevents'
    AND admissions.dischtime IS NULL
    ORDER BY chartevents.charttime DESC LIMIT 1 OFFSET 1
)
```

### <Nesting-based query>

```
SELECT
(
    SELECT chartevents.valuenum
    FROM chartevents
    WHERE chartevents.icustay_id IN (
        SELECT icustays.icustay_id
        FROM icustays
        WHERE icustays.hadm_id IN (
            SELECT admissions.hadm_id
            FROM admissions
            WHERE admissions.subject_id = 2518
            and admissions.dischtime is null
        )
    )
    AND chartevents.itemid IN (
        SELECT d_items.itemid
        FROM d_items
        WHERE d_items.label = 'admit wt'
        AND d_items.linksto = 'chartevents'
    )
    ORDER BY chartevents.charttime DESC LIMIT 1
)
-
(
    SELECT chartevents.valuenum
    FROM chartevents
    WHERE chartevents.icustay_id IN (
        SELECT icustays.icustay_id
        FROM icustays
        WHERE icustays.hadm_id IN (
            SELECT admissions.hadm_id
            FROM admissions
            WHERE admissions.subject_id = 2518
            and admissions.dischtime is null
        )
    )
    AND chartevents.itemid IN (
        SELECT d_items.itemid
        FROM d_items
        WHERE d_items.label = 'admit wt'
        AND d_items.linksto = 'chartevents'
    )
    ORDER BY chartevents.charttime DESC LIMIT 1 offset 1
)
```

Figure 5: JOIN-based and nesting-based queries.

## C.3 Comparison Between `JOIN`-based and Nesting-based Queries

Unlike most other semantic parsing datasets, the SQL queries in EHRSQL are labeled in a nested manner. Table 5 shows a comparison between queries with the naive use of JOIN and nesting. In terms of query length, JOIN-based queries are much shorter, but it is hard to follow the semantics in the queries. However, even if the length of the queries is long, T5 models are able to fully generate a long sequence of SQL (see Table 12 for qualitative results). As for execution speed, the naive use of JOIN takes almost four times slower than a nesting-based query in some cases (0.04 vs. 0.15 secs), as shown in Figure 5.

## D  Database Pre-processing Details

### D.1  Database Pre-processing Rules

- Patients aged 11 to 89 are included in the dataset.
- 1,000 patients are sampled to cover a greater number of medical events (MIMICSQL and emrKBQA use 100 patients).
- When the same type of value has multiple units of measurement, only the value with the most common unit is retained and other values are removed from the database.
- All records are lower-cased.

### D.2  Time-shifting Process

To include questions with relative time expressions, we manually time-shift each patient's hospital records. Specifically, we sample a random time point (between 2100 and 2105) to set the time of the first hospital visit. Then, we time-shift the whole patient records to the sampled time point while keeping all the record intervals the same. Additionally, we constrain the number of current patients to 10% of the total number of patients in patient sampling since any questions can be asked with relative time expressions, such as yesterday and last month.

### D.3 De-identification Process

Even though MIMIC-III and eICU are de-identified databases, they still contain patient-specific information. In our case, when one or more condition values are sampled along with a patient ID, there is a risk that the question and its paired SQL query might reveal patient-specific information. To avoid such risk, we randomly shuffle records of diagnosis, procedure, lab tests, prescription, chart events, input events, output events, microbiology tests, care units, and ward IDs across patients in the database, while keeping the time of the records the same.

## E   Template Paraphrasing Pipeline

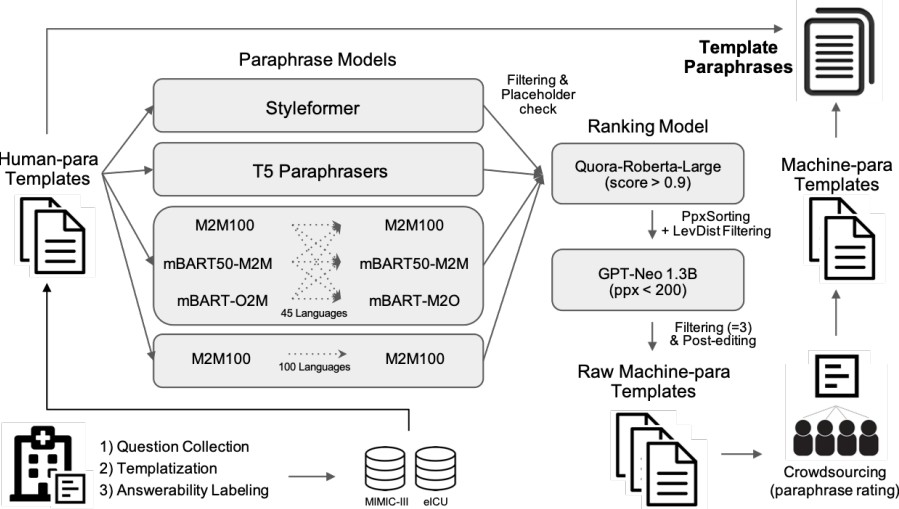

Figure 6: Template paraphrasing pipeline.

The overall template paraphrasing pipeline is illustrated in Figure 6.

## F   Data Splitting

We constrain the training, validation, and test sets to contain all question templates, but the template paraphrases do not overlap between the splits.

- Training set: We provide more text-to-SQL pairs to question templates with a greater number of slots. Specifically, question templates with fewer than three slots are assigned thirty to sixty pairs. Questions with three or more slots and fewer than five receive forty to eighty pairs; questions with five or more slots receive fifty to one hundred pairs per question template.

- Validation set: The number of pairs per question template is sampled between four and five.

- Test set: Data sampling rules are identical to those of the validation set, except that we assign higher weights on the question templates that are considered important, which are labeled in "high," "medium," "low," and "n/a" (not available) by a physician. With a score of 3 for "high," 2 for "medium," 1 for the rest, the final number of pairs in the test set is multiplied by the importance score for each question template.

- Unanswerable question: They are assigned to the validation and test sets so that each split makes up 33%.

## G    Training Details

We use pre-trained T5-base models from Hugging Face[7] for both the no schema and schema versions. Without a schema, the model is trained to translate from natural questions to SQL queries. With a schema, we additionally append schema information to the questions. The training and evaluation configurations are in Table 11. All models are trained on NVIDIA GeForce RTX 3090s.

Table 11: T5-base fine-tuning configurations.

| Training | |
| --- | --- |
| Total training step | 100,000 |
| Batch size | 32 |
| Max length | 512 |
| Optimizer | Adam |
| Learning rate | 0.001 |
| Learning rate scheduler | Fixed |
| Max gradient norm | 1.0 |
| Weight decay | 1.0 |
| Validation step | 500 |
| Evaluation | |
| Num beams | 5 |
| Repetition penalty | 1.0 |
| Length penalty | 1.0 |

## H    Qualitative Results

### H.1    SQL Generation by Question Complexity

Table 12 shows samples of generated SQL queries by the number of slots. A T5-base model trained on MIMIC-III can generate a very long sequence of SQL queries if they are seen in the training data. In most cases, the errors do not come from generating complex, long nested queries, but from failures in schema linking (identifying references of columns, tables, and condition values in natural utterances).

### H.2    SQL Generation with Different Time Expressions

Table 13 shows generated SQL samples with different time templates. The column *Seen vs. Unseen* indicates whether the exact question template (*Q_tag*) and time template (*T_tag*) combination is seen during training. Interestingly, the model can correctly generate SQL queries for both seen and unseen combinations.

### H.3    Falsely Executed and Refused Results

Table 14 and 15 show samples of falsely executed and refused questions, respectively. Falsely executed results are retrieved results of the model even though the input question is unanswerable (see Table 14). These errors are fatal mistakes that a healthcare QA system must avoid, as retrieving incorrect information may lead to wrong clinical decisions.

Table 15 shows refused results of the model. In some cases, the model might be able to generate the SQL query, but chooses not to execute it due to low confidence.

---

[7]https://huggingface.co/

Table 12: SQL generation results by complexity.

| Number of slots | Question | Real SQL | Generated SQL |
|---|---|---|---|
| 1 | what is the method for ingestion of danazol? | select distinct prescriptions.route from prescriptions where prescriptions.drug = 'danazol' | select distinct prescriptions.route from prescriptions where prescriptions.drug = 'ingestion of danazol' |
| 2 | how many patients were given temporary tracheostomy? | select count( distinct admissions.subject_id ) from admissions where admissions.hadm_id in ( select procedures_icd.hadm_id from procedures_icd where procedures_icd.icd9_code = ( select d_icd_procedures.icd9_code from d_icd_procedures where d_icd_procedures.short_title = 'temporary tracheostomy') ) | select count( distinct admissions.subject_id ) from admissions where admissions.hadm_id in ( select procedures_icd.hadm_id from procedures_icd where procedures_icd.icd9_code = ( select d_icd_procedures.icd9_code from d_icd_procedures where d_icd_procedures.short_title = 'procedures_icd') ) |
| 2 | tell me the length of the hospital stay of the last hospital stay of patient 3745. | select strftime('%J',admissions.dischtime) - strftime('%J',admissions.admittime) from admissions where admissions.subject_id = 3745 and admissions.dischtime is not null order by admissions.admittime desc limit 1 | select strftime('%J',admissions.dischtime) - strftime('%J',admissions.admittime) from admissions where admissions.subject_id = 3745 and admissions.dischtime is not null order by admissions.admittime desc limit 1 |
| 4 | tell me the percentile of mcv 97.0 in patients of the same age as patient 2127 during the last hospital visit? | select distinct t1.c1 from ( select labevents.valuenum, percent_rank() over ( order by labevents.valuenum ) as c1 from labevents where labevents.itemid in ( select d_labitems.itemid from d_labitems where d_labitems.label ='mcv') and labevents.hadm_id in ( select admissions.hadm_id from admissions where admissions.age = ( select admissions.age from admissions where admissions.subject_id = 2127 and admissions.dischtime is not null order by admissions.admittime desc limit 1 ) ) ) as t1 where t1.valuenum = 97.0 | select distinct t1.c1 from ( select labevents.valuenum, percent_rank() over ( order by labevents.valuenum ) as c1 from labevents where labevents.itemid in ( select d_labitems.itemid from d_labitems where d_labitems.label ='mcv') and labevents.hadm_id in ( select admissions.hadm_id from admissions where admissions.age = ( select admissions.age from admissions where admissions.subject_id = 2127 and admissions.dischtime is not null order by admissions.admittime desc limit 1 ) ) ) as t1 where t1.valuenum = 97.0 |
| 6 | what's the potassium change/difference of patient 99747 second measured on the last hospital visit compared to the first value measured on the last hospital visit? | select ( select labevents.valuenum from labevents where labevents.hadm_id in ( select admissions.hadm_id from admissions where admissions.subject_id = 99747 and admissions.dischtime is not null order by admissions.admittime desc limit 1 ) and labevents.itemid in ( select d_labitems.itemid from d_labitems where d_labitems.label = 'potassium') order by labevents.charttime asc limit 1 offset 1 ) - ( select labevents.valuenum from labevents where labevents.hadm_id in ( select admissions.hadm_id from admissions where admissions.subject_id = 99747 and admissions.dischtime is not null order by admissions.admittime desc limit 1 ) and labevents.itemid in ( select d_labitems.itemid from d_labitems where d_labitems.label = 'potassium') order by labevents.charttime asc limit 1 ) | select ( select labevents.valuenum from labevents where labevents.hadm_id in ( select admissions.hadm_id from admissions where admissions.subject_id = 99747 and admissions.dischtime is not null order by admissions.admittime desc limit 1 ) and labevents.itemid in ( select d_labitems.itemid from d_labitems where d_labitems.label = 'potassium') order by labevents.charttime asc limit 1 offset 1 ) - ( select labevents.valuenum from labevents where labevents.hadm_id in ( select admissions.hadm_id from admissions where admissions.subject_id = 99747 and admissions.dischtime is not null order by admissions.admittime desc limit 1 ) and labevents.itemid in ( select d_labitems.itemid from d_labitems where d_labitems.label = 'potassium') order by labevents.charttime asc limit 1 ) |

Table 13: SQL generation results with different time expressions.

| Question | Real SQL & Generated SQL | Q_tag × T_tag | Seen vs. Unseen |
|---|---|---|---|
| the first care unit of patient 46422 since 2101 is? | select transfers.careunit from transfers where transfers.hadm_id in ( select admissions.hadm_id from admissions where admissions.subject_id = 46422 ) and transfers.careunit is not null and strftime('%Y',transfers.intime) >= '2101' order by transfers.intime asc limit 1 | what was the [time_filter_exact1] careunit of patient {patient_id} [time_filter_global1]? × ('abs-year-since', '', '', 'exact-first', '') | seen |
| what is the first careunit that patient 53089 stayed on the last hospital encounter? | select transfers.careunit from transfers where transfers.hadm_id in ( select admissions.hadm_id from admissions where admissions.subject_id = 53089 and admissions.dischtime is not null order by admissions.admittime desc limit 1 ) and transfers.careunit is not null order by transfers.intime asc limit 1 | what was the [time_filter_exact1] careunit of patient {patient_id} [time_filter_global1]? × ('rel-hosp-last', '', '', 'exact-first', '') | unseen |
| count the number of patients that were prescribed aspirin ec within 2 months after having received a venous cath nec procedure until 4 years ago. | select count( distinct t1.subject_id ) from ( select admissions.subject_id, procedures_icd.charttime from procedures_icd join admissions on procedures_icd.hadm_id = admissions.hadm_id where procedures_icd.icd9_code = ( select d_icd_procedures.icd9_code from d_icd_procedures where d_icd_procedures.short_title ='venous cath nec') and datetime(procedures_icd.charttime) <= datetime('2105-12-31 23:59:00','-4 year') ) as t1 join ( select admissions.subject_id, prescriptions.startdate from prescriptions join admissions on prescriptions.hadm_id = admissions.hadm_id where prescriptions.drug = 'aspirin ec' and datetime(prescriptions.startdate) <= datetime('2105-12-31 23:59:00','-4 year') ) as t2 on t1.subject_id = t2.subject_id where t1.charttime < t2.startdate and datetime(t2.startdate) between datetime( t1.charttime) and datetime(t1.charttime,'+2 month') | count the number of patients who were prescribed {drug_name} [time_filter_within] after having received a {procedure_name} procedure [time_filter_global1]. × ('rel-year-until', '', 'within-n_month', '', '') | seen |
| how many patients were prescribed levothyroxine sodium within 2 months since 2105 after the procedure of cath base invasv ep test. | select count( distinct t1.subject_id ) from ( select admissions.subject_id, procedures_icd.charttime from procedures_icd join admissions on procedures_icd.hadm_id = admissions.hadm_id where procedures_icd.icd9_code = ( select d_icd_procedures.icd9_code from d_icd_procedures where d_icd_procedures.short_title = 'cath base invasv ep test') and strftime('%Y',procedures_icd.charttime) >= '2105' ) as t1 join ( select admissions.subject_id, prescriptions.startdate from prescriptions join admissions on prescriptions.hadm_id = admissions.hadm_id where prescriptions.drug = 'levothyroxine sodium' and strftime('%Y',prescriptions.startdate) >= '2105' ) as t2 on t1.subject_id = t2.subject_id where t1.charttime < t2.startdate and datetime(t2.startdate) between datetime( t1.charttime) and datetime(t1.charttime,'+2 month') | count the number of patients who were prescribed {drug_name} [time_filter_within] after having received a {procedure_name} procedure [time_filter_global1]. × ('abs-year-since', '', 'within-n_month', '', '') | unseen |

## H.4 Entropy Distribution of the Model Outcome

Figure 7 shows the maximum entropy values generated from T5. As the ground-truth labels (ANS: answerable; UnANS: unanswerable) indicate, the distributions of entropy values between answerable and unanswerable questions are significantly different ($< 0.001$ with the Mann-Whitney U test). Similar patterns are observed in models trained on eICU.

Table 14: Falsely executed results.

| Question | Real SQL | Generated SQL | Retrieved Answer | Comment |
|---|---|---|---|---|
| what was the duration of the packed cell transfusion procedure for patient 9566? | nan | select strftime('%J',admissions.dischtime) - strftime('%J',procedures_icd.charttime) from procedures_icd where procedures_icd.icd9_code = ( select d_icd_procedures.icd9_code from d_icd_procedures where d_icd_procedures.short_title = 'packed cell transfusion' ) ) | | Execution error |
| how long did it take to get venous cath nec for patient 31854? | nan | select strftime('%J',admissions.dischtime) - strftime('%J',admissions.admittime) from admissions where admissions.subject_id = 31854 and admissions.dischtime is not null order by admissions.admittime asc limit 1 | [(8.61111111100763,)] | Incorrect answer retrieved |

Table 15: Refused results of both correct and incorrect samples.

| Question | Real SQL | Generated SQL | Comment |
|---|---|---|---|
| how many patients were prescribed with magnesium sulfate within the same month after the treatment of cont inv mec ven <96 hrs? | select count( distinct t1.subject_id ) from ( select admissions.subject_id, procedures_icd.charttime from procedures_icd join admissions on procedures_icd.hadm_id = admissions.hadm_id where procedures_icd.icd9_code = ( select d_icd_procedures.icd9_code from d_icd_procedures where d_icd_procedures.short_title = 'cont inv mec ven <96 hrs' ) ) as t1 join ( select admissions.subject_id, prescriptions.startdate from prescriptions join admissions on prescriptions.hadm_id = admissions.hadm_id where prescriptions.drug ='magnesium sulfate') as t2 on t1.subject_id = t2.subject_id where t1.charttime < t2.startdate and datetime(t1.charttime,'start of month') = datetime(t2.startdate,'start of month') | select count( distinct t1.subject_id ) from ( select admissions.subject_id, procedures_icd.charttime from procedures_icd join admissions on procedures_icd.hadm_id = admissions.hadm_id where procedures_icd.icd9_code = ( select d_icd_procedures.icd9_code from d_icd_procedures where d_icd_procedures.short_title = 'cont inv mec ven <96 hrs' ) ) as t1 join ( select admissions.subject_id, prescriptions.startdate from prescriptions join admissions on prescriptions.hadm_id = admissions.hadm_id where prescriptions.drug ='magnesium sulfate') as t2 on t1.subject_id = t2.subject_id where t1.charttime < t2.startdate and datetime(t1.charttime,'start of month') = datetime(t2.startdate,'start of month') | Correctly generated but refused |
| how is naproxen taken? | select distinct prescriptions.route from prescriptions where prescriptions.drug = 'naproxen' | select inputevents_cv.amount from inputevents_cv where inputevents_cv.icustay_id in ( select icustays.icustay_id from icustays where icustays.hadm_id in ( select admissions.hadm_id from admissions where admissions.subject_id = naproxen ) ) and inputevents_cv.itemid in ( select d_items.itemid from d_items where d_items.label = 'niroxen'and d_items.linksto = 'inputevents_cv') | Incorrectly generated and refused |

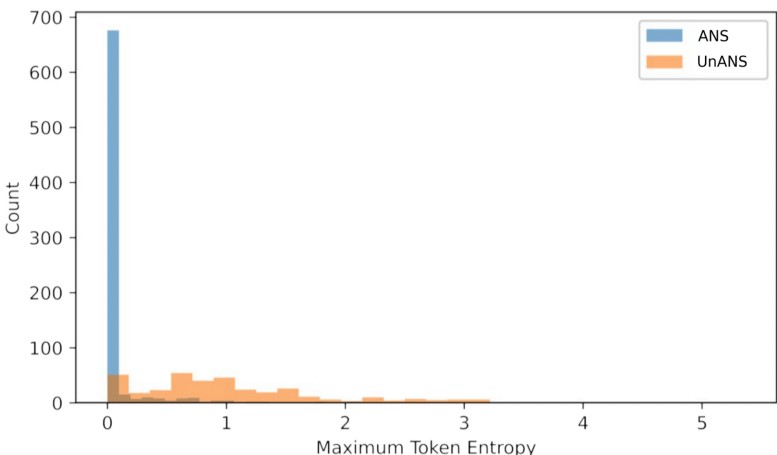

Figure 7: Distribution of entropy values generated from T5.

# I  Author statement

The authors of this paper bear all responsibility in case of violation of rights, etc. associated with the EHRSQL dataset.