# OpenReview forum: "EHRSQL: A Practical Text-to-SQL Benchmark for Electronic Health Records"
_NeurIPS.cc/2022/Track/Datasets_and_Benchmarks — NeurIPS 2022 Datasets and Benchmarks _

### Official Review · Reviewer_9Rth · 2022-07-03

**Rating:** 7
**Confidence:** 3
**Correctness:** Yes.
**Clarity:** Yes.

**Strengths:**

The paper contributes a text-to-SQL dataset using text statements that reflect real-world medical settings.

The dataset includes a large collection of 174 question templates; EHRSQL also includes an additional 56 time-based templates, resulting in significantly more time-based text-to-SQL pairs than existing datasets.

EHRSQL contains "unanswerable" questions, further simulating real-world conditions, something that no existing dataset contains. Based on the addition of these instances, the authors introduce the task of "trustworthy semantic parsing", in which one must consider the validity of the model's answer as well as its ability to refuse questions beyond its capability.

The authors provide paraphrased question templates to add linguistic variety to the dataset. These paraphrased questions are filtered using human evaluation.

The paper is well-written and easy to read and follow.

The dataset has the potential to positively impact an important area of research.

**Weaknesses:**

One of my main concerns is the comparison of EHRSQL to emrKBQA, since emrKBQA seems to be a similar but much larger dataset than EHRSQL (940K vs. 24K text-to-SQL pairs for emrKBQA and EHRSQL, respectively), potentially limiting the impact of the proposed EHRSQL dataset.

How extensible is the EHRSQL dataset? It was not clear to me how one could extend this dataset, and if so, how easy or difficult that process would be.

62% of the responses are from the nursing staff vs. 22% for physicians. Are there observable differences between the medical statements made by these two medical groups?

MINOR WEAKNESSES

There are multiple grammatical errors, I recommend using a service like Grammarly to fix these issues.

line 156: No space between "...revisions.Space..."

**Additional Feedback:**

No.

**Documentation:**

Yes.

**Ethics:**

No.

**Relation To Prior Work:**

Yes.

**Summary And Contributions:**

The paper proposes EHRSQL, a text-to-SQL dataset for semantic parsing of electronic health records (EHRs). The authors collected 1,742 utterances from 222 medical professionals from the Konyang University hospital to build 174 question templates and 56 time templates;
statements that were ambiguous, required external knowledge, or were outside the scope of the schema were filtered out and used as "unanswerable" questions to simulate potential queries that may occur in real-world medical settings. SQL annotation is then performed for each template; then a sampling process is used to generate all text-to-SQL pairs.

Human and machine-generated paraphrases of the question templates were also included to add linguistic variety to the dataset. Overall, EHRSQL contains 24,211 text-to-SQL pairs linking two medical databases (MIMIC-III and eICU); EHRSQL includes multi-table queries, time queries, and unanswerable questions to further bridge the gap between existing text-to-SQL datasets and real-world medical query settings.

---

> ### Author Response · Authors · 2022-08-15
> **Response to Reviewer 9Rth**
>
> Thank you for the valuable comments and suggestions. Please kindly find the responses below.
>
> **Q: One of my main concerns is the comparison of EHRSQL to emrKBQA, since emrKBQA seems to be a similar but much larger dataset than EHRSQL (940K vs. 24K text-to-SQL pairs for emrKBQA and EHRSQL, respectively), potentially limiting the impact of the proposed EHRSQL dataset.**
>
> We believe our dataset is different from emrKBQA in several aspects. In addition to ours being the first semantic parsing dataset for eICU as well as MIMIC-III, emrKBQA uses the question templates proposed in emrQA, which are based on outpatient clinical notes, and this limits the compatibility with ICU records [1] like MIMIC and eICU. Our dataset, on the other hand, does not rely on a particular group of patients, as we asked the poll taker to write down any questions they look for in structured information in the EHR, regardless of outpatient, inpatient, and ICU. After curating them for our semantic parsing task, we have a solid set of questions that covers a wide range of concepts in the hospital and are compatible with either or both MIMIC and eICU databases (the complete list is reported in the supplementary material).
>
> Moreover, we believe being large-scale in emrKBQA does not necessarily imply diversity in data. [2] claims that using only a subset (5%-20%) of emrQA achieves almost the same performance as using the entire data. Since emrKBQA is based on emrQA, we reasonably suspect that the same pattern can also be observed in emrKBQA. Reviewer SQY6 also informs that "analysis of the emrKBQA dataset showed that training a model on only 5% of the data was equivalent to training the model on 100% of the data" (this claim is unverifiable, since emrKBQA has never been released to the public, and the paper does not mention this 5% issue).
>
> In our case, we also tried to increase the number of data by random condition value sampling, but naively increasing the dataset did not significantly improve performance. Also, our value sampling strategy requires joint sampling of values when multiple slots exist in one question. The discrepancy in the number of slots may cause an unbalanced number of sampled data per question template, which may overlook some important question templates. If anyone, however, wants to increase the size of the training data, one can use the "value" data field (i.e., stores condition values) to replace the original value with a randomly sampled value (but this might not always be executable).
>
> **Q: How extensible is the EHRSQL dataset? It was not clear to me how one could extend this dataset, and if so, how easy or difficult that process would be.**
>
> Since we include a separate data field for storing condition values, one can easily extend the training data size by randomly sampling values from the database. Also, according to the suggestion by Reviewer whS9, one could extend this dataset to more interactive settings such as semantic parsing in context or iterative query refinement. The seed questions in EHRSQL can serve as a great starting point for related healthcare question answering tasks.
>
>
> **Q: 62% of the responses are from the nursing staff vs. 22% for physicians. Are there observable differences between the medical statements made by these two medical groups?**
>
> We observed some differences in questions collected from different professionals. For example, questions collected from nurses often focus on current patient-specific medical events (e.g., time of I&O and vital signs) and verifying a patient’s past medical check-up values. Physicians focus on patient-specific medical events and sometimes longitudinal cohort-based questions (e.g., counting the number of patients within a certain age range who got some medication after being diagnosed with some disease). Insurance review and marketing staff are interested in knowing the total hospital cost of a patient and the number of procedures conducted in the hospital.
>
> **Q: There are multiple grammatical errors, I recommend using a service like Grammarly to fix these issues.**
>
> We have fixed several grammatical errors and plan to proofread until the final draft submission.
>
> **Reference**
>
> [1] Raghavan, Preethi, et al. "emrkbqa: A clinical knowledge-base question answering dataset." Proceedings of the 20th Workshop on Biomedical Language Processing. 2021.
>
> [2] Yue, Xiang, et al. "Clinical Reading Comprehension: A Thorough Analysis of the emrQA Dataset." Proceedings of the 58th Annual Meeting of the Association for Computational Linguistics. 2020.

---

### Official Review · Reviewer_whS9 · 2022-07-20
**A more realistic Text-to-SQL dataset for medical domain applications.**

**Rating:** 8
**Confidence:** 5

**Strengths:**

At the time of writing, this publicly available dataset is not the largest one available for this task and domain in terms of size. But it proposes numerous improvements, which are particularly important for real-world usage of the tools. The authors invest a lot of time and effort in the evaluation and detection of the trustworthiness of the systems, and none of the previously released datasets really tackle the issue of unanswerable questions in the same way that this team does. Compare to previous works, they invest a lot of human labor to come up with much more realistic data : from the technical side point by offering more and bigger SQL tables and from the linguistic side point, since the utterances are way closer to real natural language, if we compare it directly to the MIMICSQL dataset. Adding one more layer of de-identification during the generation process, by randomly shuffling values across patients to make the values untraceable and be able to publicly release the dataset without any more requirement, is something not negligible. It's also reinforcing this idea of ​​privacy protection in machine learning applied to healthcare. The semi-automatic paraphrasing pipeline seem to provide good quality outputs and is sufficiently explained to be replicated by ourselves using the code you are providing freely on the GitHub repository of the project.

**Weaknesses:**

The emrKBQA dataset contains only twice as many utterances after the collection and cleaning process, but have 39 time more utterances than yours after data-augmentation / paraphrasing. You put a lot of effort to obtains high quality paraphrasing, which is a very good point, but you haven't invested time to augment the data by filling up slots with "random" medications and diseases to artificially obtains more data.

If I take for example the template "what is the intake method of {drug_name}?", we got the following paraphrase variations :

- what are the methods of consumption of potassium chloride 20 meq/50 ml iv piggy back 50 ml bag?
- tell me the intake method for potassium chloride tabs?
- what is the ingesting method of levothyroxine?

But none of them are duplicated sentence with variations only for medications and diseases. Unfortunately, you didn't do it, however the corpus could have easily grown as a result. It also could be interesting to learn about it if you have tried this strategy, whatever the results could have been.

I also don't understand why when I am counting the number of elements in the train.json and valid.json files for MIMIC-III and eICU, I only got 20,837 elements when you are advertising 24K examples in the table 4 ? Is it a typo ? Or the GitHub repository isn't up-to-date ?

**Additional Feedback:**

A tremendous amount of work was put in place to collect such utterances and extend current dataset capabilities. They put many efforts into understanding errors in the state-of-the-art T5 model by integrating unanswerable questions during the evaluation phase. Lot of works as been invested in the paraphrase generation system to produce high quality variations and all their works has been generously open-sourced on GitHub to encourage reproducibility.

It will also be very interesting to get our hand on the human annotations used to rate the quality of the paraphrases, since it can be useful to set up an automated system for healthcare.

**Clarity:**

Well organized, written with a good english. The only thing I would like to have is a detailed description of the text-to-SQL task itself, since the paper have to be open to a broad range of peoples, sometimes not familiar with the task. It was a little confusing at the start for me and require reading the paper a few times to capture the meaning of it. A simple diagram of the followed pipeline at the beginning of the paper could dramatically improve readability of it.

**Correctness:**

Both metrics : execution accuracy (EX) and exact string match (ESM), are generally used to evaluate the task for similar datasets (MIMIC-III and eICU). Since the authors have also put their effort in evaluating unanswerable questions, they introduce another metric called false-negative rate (FNR). Experiments in the other hand are based on three architectures : GAP, T5 and T5+Schema which represent correctly the approaches that could be used to perform the task at a state-of-the-art level.

**Documentation:**

For the dataset collection and organization, it's very clear. For the availability, the authors put their effort in releasing the dataset and the code on GitHub with a nice description to encourage reproducibility. The licensing is available on the GitHub repository but no particular license is defined in the paper for the extension you are proposed, it will be simpler for the papers to follow and users interested in the project to get this information in the paper and not digging to get the information.

To draw even more people to the task and enable better growth from the performance's standpoint, it will be awesome to have access to the corpus on other platforms like Zenodo. It's also crucial to include contact information on the GitHub page, and the update schedule is unclear.

From the ethical and responsible point of view, nothing to complain about this. The user's anonymity is fully preserved.

The approaches used during the benchmarks section have clear parameters and allow a good reproducibility.

**Ethics:**

Does the dataset use features or label information about individual names?
> No.

Did people provide their consent on the collection of such data?
> Yes.

Could the use of the data be degrading or embarrassing for some people?
> No, the data is based on MIMIC-III dataset which contains sensitive information about patients, but it has been de-identificated to make the real patient name untraceable by using pseudonymization. The access to MIMIC-III dataset require credentialed access to PhysioNet. The current dataset also performed a random shuffling of values across patients to make the values like diseases untraceable and be able to publicly release the dataset without any more requirement.

Contain information that could be deduced about individuals that they have not consented to share ?
> No.

Encode, contain, or potentially exacerbate bias against people of a certain gender, race, sexuality, or who have other protected characteristics ?
> No.

Contain human subject experimentation and whether it has been reviewed and approved by a relevant oversight board ?
> No.

Have been discredited by the creators ?
> No.

**Relation To Prior Work:**

Absolutely clear, the authors take their time to describe the datasets available for the task and the domain during the tables 4. However, authors could also refer to the corpus built on top of Spider like SParC (https://arxiv.org/abs/1906.02285) and SPLASH (https://www.aclweb.org/anthology/2020.acl-main.187/) to demonstrate that some other similar initiatives has also been experimented on datasets other than MIMIC-III.

**Summary And Contributions:**

The authors propose an extension of the capabilities of the well-known Text-To-SQL datasets MIMIC-III and eICU by introducing 24,000 more realistic utterances, manually collected from the 222 participants of the Konyang University hospital and obtained using paraphrasing augmentation on the base 1,742 utterances.

The collection of real-world intentions is the dataset's major contribution. Then, the dataset is structured around the concept of trustworthiness, which was generally forgotten is the previous works despite the fact that it's a huge concern in medical applications. Finally, they employ paraphrase to do cutting-edge data augmentation and increase the dataset's possibilities.

The authors also evaluate this dataset performance using off the shelves state-of-the-art semantic parsing models and compare them to more general-purpose state-of-the-art sequence to sequence models in various scenarios for highlighting their limits in more realistic uses-cases.

---

> ### Author Response · Authors · 2022-08-15
> **Response to Reviewer whS9 (1/2)**
>
> Thank you for the valuable comments and suggestions. Please kindly find the responses below.
>
> **Q: The emrKBQA dataset contains only twice as many utterances after the collection and cleaning process, but have 39 time more utterances than yours after data-augmentation / paraphrasing. You put a lot of effort to obtains high quality paraphrasing, which is a very good point, but you haven't invested time to augment the data by filling up slots with "random" medications and diseases to artificially obtains more data.**
>
> According to [1], using only a subset (5%-20%) of emrQA achieves almost the same performance compared to the model trained on the entire dataset. Since the questions in emrKBQA are based on emrQA, we reasonably suspect that a similar pattern may also happen in the emrKBQA dataset. This finding also aligns with Reviewer SQY6’s comment, "analysis of the emrKBQA dataset showed that training a model on only 5% of the data was equivalent to training the model on 100% of the data" (keep in mind that this claim is unverifiable, since emrKBQA has never been released to the public, and the paper never mentions about this 5% issue). Therefore, we believe being large-scale in emrKBQA does not necessarily imply diversity in data.
>
> In our case, we similarly observed that naively increasing the number of data to the order of 100K via random condition value sampling only gave us a 3% performance increase. Also, our value sampling process requires a joint value sampling strategy when multiple slots exist in one question. This discrepancy in the number of slot values quickly results in an unbalanced generation of samples per question template. For these reasons, we set our data size to be 24K with a complexity-adjusted number of data per question template. Furthermore, anyone can increase the size of the training data by using the "value" data field (i.e., stores condition values) to replace the original value with a randomly sampled value (but this might not always be executable).
>
> **Q: I also don't understand why when I am counting the number of elements in the train.json and valid.json files for MIMIC-III and eICU, I only got 20,837 elements when you are advertising 24K examples in the table 4 ? Is it a typo ? Or the GitHub repository isn't up-to-date ?**
>
> You are correct that the number of data in train and validation splits is 21K. Other 3K data are saved for a hidden test set. Below is the exact number of data instances, and we have added more details in Section 4.1.
>
> |                 | Train         | Valid     | Test     |     All   |
> | :--------- | :---------- | :------ | :------ | :------ |
> | MIMIC-III | 9,323       | 1,122   | 1,787  | 12,232 |
> | eICU        | 9,270       | 1,117   | 1,792  | 12,179 |
> | All            | 18,593     | 2,239   | 3,579  | 24,411 |
>
>
> **Q: The only thing I would like to have is a detailed description of the text-to-SQL task itself, since the paper have to be open to a broad range of peoples, sometimes not familiar with the task. It was a little confusing at the start for me and require reading the paper a few times to capture the meaning of it. A simple diagram of the followed pipeline at the beginning of the paper could dramatically improve readability of it.**
>
> Thank you for the suggestion. We will add a diagram that illustrates the overview of our new task (https://drive.google.com/file/d/19FyMsCLUxbmnahgySlLrCp7xrzuxlmxn/view?usp=sharing). Due to the space limit of 9 pages, we plan to include it in the next version of the paper.
>
> **Q: The authors could also refer to the corpus built on top of Spider like SParC (https://arxiv.org/abs/1906.02285) and SPLASH (https://www.aclweb.org/anthology/2020.acl-main.187/) to demonstrate that some other similar initiatives has also been experimented on datasets other than MIMIC-III.**
>
> Thank you for the suggestion. We agree that SParC and SPLASH are great ways to expand our research. We have mentioned the initiatives as one of the research directions in the conclusion.

---

> > ### Author Response · Authors · 2022-08-15
> > **Response to Reviewer whS9 (2/2)**
> >
> > **Q: The licensing is available on the GitHub repository but no particular license is defined in the paper for the extension you are proposed, it will be simpler for the papers to follow and users interested in the project to get this information in the paper and not digging to get the information. / To draw even more people to the task and enable better growth from the performance's standpoint, it will be awesome to have access to the corpus on other platforms like Zenodo. It's also crucial to include contact information on the GitHub page, and the update schedule is unclear.**
> >
> > Thank you for the suggestion. We have added contact information on the Github page and a link to the paper which has more details about uses, distribution, and maintenance in Datasheets for Datasets. We also plan to release our data on other data-sharing platforms such as PhysioNet and Zenodo. Regarding the license, we are publishing our dataset under a CC license instead of the MIT License, following the NeurIPS guideline.
> >
> > **Reference**
> >
> > [1] Yue, Xiang, et al. "Clinical Reading Comprehension: A Thorough Analysis of the emrQA Dataset." Proceedings of the 58th Annual Meeting of the Association for Computational Linguistics. 2020.

---

### Official Review · Reviewer_SQY6 · 2022-07-24
**EHRSQL: A Practical Text-to-SQL Benchmark for Electronic Health Records**

**Rating:** 7
**Confidence:** 4

**Strengths:**

The work presented employed a poll to source questions, and applied more advanced models as compared to emrKBQA. It offers additional slot-filled templates and paraphrases on top of emrKBQA dataset. And although the methodology presented is not new, the modeling, specifically the T5 model, is solid. A strong baseline model was used to evaluate the test set.

**Weaknesses:**

The authors recruited non-medical annotators to determine whether or not the paraphrases generated are similar or distinct from the original questions. However, questions may mean different to clinicians versus non-experts.
The performance of a zero-shot setting is slightly misleading. An ablation study is more appropriate to truly assess zero-shot learning as slot filling often only leads to redundant training data. For example, analysis of the emrKBQA dataset showed that training a model on only 5% of the data was equivalent to training the model on 100% of the data.
How does the model perform when it encounters paraphrases it has never seen before? A paraphrase level split of performance should be presented. The breakdown of the results is extremely weak. Statistics of where the model is wrong is needed, i.e., percent of time the model fails on X).
Finally, what is the point of paraphrase generation if there is a need for three annotators to assess them?

**Additional Feedback:**

Although the submission provides a unique dataset, but it is currently unclear how much it differs from the emrKBQA dataset. The questions are presented in the Appendix but no analysis was performed to compare with emrKBQA.

**Clarity:**

The main claim is that the EHRSQL dataset is more diverse, but this is not substantiated. There are significant gaps in the description of the data collection. For example, the slot filling of the templates is lacking in detail. Much more analysis of the questions generated is needed.
●	Why are there unanswerable questions in the EHRSQL if the instruction was specifically not to ask questions that require “external knowledge, ambiguous or qualitative statements, or asking for reasons behind some clinical decisions”? How many of the 1,742 questions fall under this category?
●	How frequently did medical experts ask about “groups of patients”?
●	How were “conditions” chosen in stage 3 of slot filling? by ICD-9 codes?
●	What is the point of the de-identification section? The patient IDs are already random.




**Correctness:**

While the questions generated were technically written by medical experts, the claim that they accurately “reflect user interests” or are “naturally-occuring” is misleading, as the questions were taken from a poll, rather than from a real clinical setting such as while rounding. Further, by templatizing and then slot-filling these questions, the distribution of the questions changes. This leads to questions that are structurally realistic, but may not be relevant. Testing on this data likely gives researchers a false sense of confidence, especially if the questions are not broken down into different categories.



**Documentation:**

The slot filling of the templates is lacking in details. The following aspects of data collection and analysis need more elaboration:
●	Why are there only 230 templates? What happened to the rest of the questions? Were they duplicates?
●	How were questions that were “bad” (from the 1700+) filtered out? It just says “we filtered out utterances that did not meet the criteria”.
●	What criteria were used to determine if a paraphrase was assigned a pass/fail?
●	How frequently did these annotators agree with one another? What training were they given? How long did it take to score each question? How much money per hour were they paid (should be included in appendix)?
●	By filtering down the 1700+ questions to 230 templates, did the authors see a different distribution of questions?
●	Were paraphrases added to the test set?
●	How many questions were scored on average? It would be helpful to know what percent of questions were accepted by the annotators.

**Relation To Prior Work:**

The authors correctly pointed out that the emrKBQA dataset is not diverse, as it mostly asks about test results. However, there is no analysis to demonstrate that the dataset presented is more diverse than the emrKBQA dataset. A breakdown of the types of questions asked is necessary in order to support the claim that the questions generated are more diverse. Moreover, there was no breakdown of model performance with respect to the different types of questions.

**Summary And Contributions:**

Similar to emrKBQA, this work created a repository of medical questions framed around the MIMIC and eICU-CRD datasets, and assembled them into templates. The authors generated a total of 24,000 queries. To increase the diversity of the questions, the authors employed a collection of models (i.e., T5, Roberta, GPT-Neo) to generate, on average, 21 paraphrases per question template. The authors then trained a T5 model with schema serialization on their data and reported an execution accuracy of around 86% on both the MIMIC and eICU-CRD splits.
The claim of presenting a corpus of realistic questions is misleading, as the questions were taken from a poll, rather than from a real clinical setting such as while rounding. . The authors failed to provide issues with slot-filling.
Further, more comparison to prior work, i.e. emrKBQA, is necessary in order to ground the data and the results presented.

---

> ### Author Response · Authors · 2022-08-15
> **Response to Reviewer SQY6 (1/4)**
>
> Thank you for the time and effort spent in carefully reviewing our work. Please kindly find the responses below.
>
> **Q: The authors recruited non-medical annotators to determine whether or not the paraphrases generated are similar or distinct from the original questions. However, questions may mean different to clinicians versus non-experts.**
>
> As described in Section 3.3 and Appendix E, condition values (e.g., disease names, medication names, etc.) in the paraphrase samples were all replaced with short simple placeholders when given to the crowdsource workers (e.g., all disease names were replaced with "pneumonia"). Therefore we believe the workers would have been sufficiently capable of distinguishing a good paraphrase from a bad one. Furthermore, in the initial annotation rounds, the crowdsourcing company and our group communicated several times to clarify which qualify as good paraphrases and which do not, and their questions were not related to clinical terms or meanings, but always related to common English errors (e.g., Should "How old is the patient?" and "What are the age of the patient?" be seen as good paraphrases, even though "are" is a wrong choice of verb?).
>
> **Q: The performance of a zero-shot setting is slightly misleading. An ablation study is more appropriate to truly assess zero-shot learning as slot filling often only leads to redundant training data. For example, analysis of the emrKBQA dataset showed that training a model on only 5% of the data was equivalent to training the model on 100% of the data. How does the model perform when it encounters paraphrases it has never seen before? A paraphrase level split of performance should be presented.**
>
> Please note that the zero-shot experiment has nothing to do with the training data. In the zero-shot experiment, the model was trained on the Spider dataset [1], then tested on the test samples from MIMICSQL [2] and ours (both MIMIC-III and eICU). We conducted the zero-shot experiment to demonstrate that, compared to MIMICSQL, our dataset is more complex, and has a more distinguished characteristic from the general domain semantic parsing datasets.
>
> As for paraphrase level splitting, we completely agree, and that is why our dataset was constructed exactly in that fashion: in Appendix F, we explicitly mention that the template paraphrases do not overlap between the train/validation/test splits. Therefore, all performance results in Table 5 are indeed "how the model performs when it encounters paraphrases it has never seen before".
>
> Lastly, unless one is directly involved in the emrKBQA project, there is no way of knowing that the "analysis of the emrKBQA dataset showed that training a model on only 5% of the data was equivalent to training the model on 100% of the data", because the dataset has never been released to the public yet, and the paper [3] never mentions about 5% of the training data being sufficient. Therefore we treat this as an unverifiable claim (but very feasible, because emrKBQA uses a subset of question templates from emrQA, which also has been shown by Yue et al. [4] to have constructed more-than-necessary training samples). Please note that the reason we chose our dataset size to 24K is exactly because generating more samples only made a negligible performance difference.
>
> **Q: The breakdown of the results is extremely weak. Statistics of where the model is wrong is needed, i.e., percent of time the model fails on X.**
>
> We report qualitative results of T5 models (Table 12, 13, 14, 15) for different scenarios (e.g., question complexity, different time expression, falsely executed, and refused results) instead of how much each error takes up in percentage. As our task adds more complexity to text-to-SQL, involving generalizing to unseen time and question combinations and handling model uncertainty in predictions, we consider each scenario equally important and attempt to avoid any misconceptions caused by the data distribution. We have added the performance of T5 models for each question template in Section H.5 in the supplementary material.
>
> **Q: What is the point of paraphrase generation if there is a need for three annotators to assess them?**
>
> With a fixed budget, one can either ask the crowdsource workers to manually write paraphrases given the initial seeds, or one can generate paraphrases with the help of large language models and ask the workers to check whether the machine-generated paraphrases are valid. Clearly, the cognitive load is lighter for the latter option, which enables a larger number of paraphrased samples in the end. We hope that any researchers with a limited budget may find our strategy helpful.

---

> > ### Author Response · Authors · 2022-08-15
> > **Response to Reviewer SQY6 (2/4)**
> >
> > **Q: While the questions were technically written by medical experts, the claim that they accurately “reflect user interests” or are “naturally-occuring” is misleading, as the questions were taken from a poll, rather than from a real clinical setting such as while rounding.**
> >
> > To reflect the interests of different professionals, we specifically asked them what questions they would ask an AI speaker for information they frequently find in the structured EHR during daily work. Since the questions are what the poll takers claim to look for in the database, we treat that the questions reflect the needs of the different professionals, and the structured information to be retrieved is grounded on the actual use cases. Regarding "naturally occurring," we admit that this might lead to a misunderstanding of our data collection process, so we have changed the phrase to "realistic" instead.
> >
> > **Q: Further, by templatizing and then slot-filling these questions, the distribution of the questions changes. This leads to questions that are structurally realistic, but may not be relevant. Testing on this data likely gives researchers a false sense of confidence, especially if the questions are not broken down into different categories.**
> >
> > Thank you for acknowledging the structural realism of our dataset. As you mentioned, slot-filled-values could affect further realisticness. But in a typical semantic parsing task, the condition values are given less weight than the structure of the query (if kept the same), as they can be copy-pasted from the input question. Furthermore, our templates are created in a way to maximally preserve the original intention of the poll-takers, such that each slot in a template has a specific category (e.g., a slot can be filled only with a drug). Therefore, we believe that the intentions of the poll takers remain kept.
> >
> >
> > **Q: The main claim is that the EHRSQL dataset is more diverse, but this is not substantiated. / The authors correctly pointed out that the emrKBQA dataset is not diverse, as it mostly asks about test results. However, there is no analysis to demonstrate that the dataset presented is more diverse than the emrKBQA dataset. A breakdown of the types of questions asked is necessary in order to support the claim that the questions generated are more diverse.**
> >
> > Unfortunately, the authors of emrKBQA have not released the actual dataset, so we could not compare EHRSQL with emrKBQA side by side. However, according to the paper, emrKBQA uses the questions from emrQA, which are created for answering outpatient-related questions. As a result, the source of the questions in emrKBQA is not best suitable for question answering with ICU records [3]. Our dataset, on the other hand, does not rely on a particular group of patients, as the poll takers were asked to write down any questions they look for in structured information in the EHR, regardless of outpatient, inpatient, and ICU.
> >
> > As a result, after careful templatization, the questions cover a wide range of concepts in the hospital (e.g., calculating average costs, survival rates, and longitudinal statistics) and are the first semantic parsing dataset compatible with both eICU and MIMIC-III. Please refer to the list of 174 question-SQL pairs in the supplementary material (emrKBQA has 52 logical forms with mostly retrieving test results).
> >
> >
> > **Q: Why are there unanswerable questions in the EHRSQL if the instruction was specifically not to ask questions that require “external knowledge, ambiguous or qualitative statements, or asking for reasons behind some clinical decisions”? / How many of the 1,742 questions fall under this category?**
> >
> > Even after providing instructions on what kind of questions can and cannot be asked, unanswerable utterances were still observed because not all poll takers were familiar with the concept of ambiguity or external knowledge in the context of semantic parsing. In addition, another big reason was that they were not given information about MIMIC-III and eICU database structures, which became one of the crucial factors in collecting real-world utterances in the semantic parsing literature [5, 6]. Among 1,742 collected questions, half of the questions were considered to be not answerable in both MIMIC-III and eICU.
> >
> > **Q: What is the point of the de-identification section? The patient IDs are already random.**
> >
> > As privacy is a huge concern in healthcare, we want to ensure that patient-specific information is completely removed. Although patient IDs are already de-identified, one could guess a patient suffered from disease A and got medication B if the condition values (disease A and medication B) in the utterance are sampled from a real patient (with random patient ID). To alleviate such cases and release the text-to-SQL pairs without risking privacy, we further remove patient-specific information to make patient-specific information completely untraceable.

---

> > > ### Author Response · Authors · 2022-08-15
> > > **Response to Reviewer SQY6 (3/4)**
> > >
> > > **Q: How frequently did medical experts ask about “groups of patients”?**
> > >
> > > After the poll, the answerable questions consisted of 90% individual-level questions and 9% group-level questions. However, after templatization, 57 question templates are considered group-level, and the numbers of individual-level and non-patient questions are 111 and 6, respectively.
> > >
> > >
> > > **Q: How were “conditions” chosen in stage 3 of slot filling? by ICD-9 codes?**
> > >
> > > They are the curly bracket slots in question templates, and they can be replaced with values sampled from the columns listed in Table 10. To better demonstrate which condition value slots map to which columns, we have added the schema mapping in the supplementary material.
> > >
> > >
> > > **Q: Why are there only 230 templates? What happened to the rest of the questions? Were they duplicates?**
> > >
> > > We observed many duplicate utterances in the poll result. Following the steps explained in Section 3.1.1, we first grouped the questions into three patient-based categories: a single patient, a group of patients, and no patient. Then, we merged duplicate utterances into the same question templates. Words that can be replaced with values in the database became slots. We also templatize unanswerable questions that could be merged and consider them the questions that the model should not answer to evaluate model trustworthiness.
> > >
> > > **Q: How were questions that were “bad” (from the 1700+) filtered out? It just says “we filtered out utterances that did not meet the criteria”.**
> > >
> > > As mentioned in Section 3.1.1, the criteria we decide for accessing unanswerable questions include ambiguous statements, those that require external knowledge, or those that go beyond the database schema. Below are some of the examples we consider unanswerable:
> > > - What are the 10 most recent papers on {diagnosis_name}? (the word “recent” is ambiguous; publication information is beyond the database schema)
> > > - Tell me the name of the diagnosis that patient {patient_id} received in the other department. (department information is beyond the database schema)
> > > - What is the protocol of anticancer drugs? (protocol of medication administration is an example of external medical knowledge)
> > >
> > > Depending on the level of ambiguity, some questions were occasionally modified to be answerable, for example, by removing ambiguous words.
> > >
> > >
> > > **Q: What criteria were used to determine if a paraphrase was assigned a pass/fail? / What training were they given?**
> > >
> > > We provided instructions on how to label a pass or fail. The basic rule was whether two sentences mean the same thing. More specifically, we asked them to pay special attention to the target concept (e.g., what information is asked?) and a way of combining the information (e.g., how many, what was, and list all, etc.), which all determine the final SELECT clause in SQL. Small grammatical mistakes and inconsistencies in verb tenses were considered acceptable. For condition values, we ensured every paraphrased question includes the same condition values as the original sentences.
> > >
> > >
> > > **Q: How frequently did these annotators agree with one another? / How long did it take to score each question? How much money per hour were they paid (should be included in appendix)? / How many questions were scored on average? It would be helpful to know what percent of questions were accepted by the annotators.**
> > >
> > > Among 25,140 machine-generated paraphrases, the average number of pass votes was 1.64. Below is the summary of the scores given by three annotators.
> > >
> > > | Number of pass votes  | 0 vote          | 1 vote          | 2 vote           | 3 vote          |
> > > | :------------------------- | :------------- | :------------- | :------------- | :------------- |
> > > | 25,140 (100%)             | 6,283 (25%) | 4,836 (19%) | 5,550 (22%) | 8,471 (34%) |
> > >
> > > We only selected the ones with unanimous votes (3 votes) as our final paraphrases. Annotators were paid hourly ($18 per hour), and the average time it took to score each question was approximately 30 seconds.
> > >
> > > **Q: By filtering down the 1700+ questions to 230 templates, did the authors see a different distribution of questions?**
> > >
> > > Since we removed any duplicate questions, the notion of frequently asked questions became no longer available. However, we alleviate this concern by adopting importance scores on every question template, labeled by two physicians.
> > >
> > > **Q: Were paraphrases added to the test set?**
> > >
> > > Yes, but we ensure the same template paraphrases are not present in different data splits.

---

> > > > ### Author Response · Authors · 2022-08-15
> > > > **Response to Reviewer SQY6 (4/4)**
> > > >
> > > > **Reference**
> > > >
> > > > [1] Yu, Tao, et al. "Spider: A large-scale human-labeled dataset for complex and cross-domain semantic parsing and text-to-SQL task." 2018 Conference on Empirical Methods in Natural Language Processing, EMNLP 2018. Association for Computational Linguistics, 2020.
> > > >
> > > > [2] Wang, Ping, et al. "Text-to-SQL generation for question answering on electronic medical records." Proceedings of The Web Conference 2020. 2020.
> > > >
> > > > [3] Raghavan, Preethi, et al. "emrkbqa: A clinical knowledge-base question answering dataset." Proceedings of the 20th Workshop on Biomedical Language Processing. 2021.
> > > >
> > > > [4] Yue, Xiang, et al. "Clinical Reading Comprehension: A Thorough Analysis of the emrQA Dataset." Proceedings of the 58th Annual Meeting of the Association for Computational Linguistics. 2020.
> > > >
> > > > [5] Lee, Chia-Hsuan, et al. "KaggleDBQA: Realistic Evaluation of Text-to-SQL Parsers." Proceedings of the 59th Annual Meeting of the Association for Computational Linguistics and the 11th International Joint Conference on Natural Language Processing (Volume 1: Long Papers). 2021.
> > > >
> > > > [6] Hazoom, Moshe, et al. "Text-to-SQL in the Wild: A Naturally-Occurring Dataset Based on Stack Exchange Data." Proceedings of the 1st Workshop on Natural Language Processing for Programming (NLP4Prog 2021). 2021.

---

### Official Review · Reviewer_awo2 · 2022-07-27

**Rating:** 10
**Confidence:** 5
**Clarity:** The paper is very well written and is…

**Strengths:**

The following are the strengths of the papers:
1. The dataset would be a promising contribution to the EHR question-answering community. It is the first text-to-SQL dataset for the eICU database. Since they address real-world questions, so they should be quite relevant to the EHR research community.
2. The dataset will help to test trustworthiness of QA systems. Also, the dataset is time-shifted to keep the time span of the two EHR databases (i.e. MIMIC-III and eICU) realistic.
3. The effort made in the paraphrase generation module was very impressive. The steps involve using human paraphrasing along with use of T5 paraphrasers.
4. There does not seem to be any ethical and social implications.



**Weaknesses:**

Some of the weaknesses of this paper are as follows:
1. Section 3.2 mentions the concept of "cost sampling procedure". The authors need to provide a more elaborate and clear explanation of how the cost is computed from the MIMIC-III database. Some examples should be used to explain the idea.
2. Even though the paper has experimented with models like T5 and GAP for the EHRSQL dataset, the authors should have also evaluated its performance on the "TREQS model" (TRANSLATE-EDIT MODEL FOR QUESTION-TO-SQL QUERY GENERATION) which was used originally for the MIMICSQL dataset.  Or at least, the authors should have mentioned the reason for not evaluating using the TREQS model.


**Additional Feedback:**

Even though there are a few minute weaknesses of the paper, they can be overlooked since it is a significantly useful dataset for the EHR question-answering community.

**Correctness:**

The authors have provided a well-detailed description of the dataset creation. They have listed all the templates in the Appendix section. The claims in the submission seem to be correct. The paper clearly mentions the models used to evaluate the dataset.

**Documentation:**

There is sufficient data about data collection, availability, ethical use, and maintenance. The dataset and the code are made publicly available on their GitHub repository. Licensing and maintenance plans are discussed.

**Ethics:**

There does not seem to be any ethical concern.

**Relation To Prior Work:**

The paper was successful in discussing its contributions and its relevance in comparison to previous works.

**Summary And Contributions:**

This paper has introduced a novel text-to-SQL dataset (EHRSQL) for two publicly available EHRs- MIMIC-III and eICU. The questions are collected using a poll conducted at a hospital. They address real-world questions. The questions include time-sensitive questions and also include unanswerable questions from the poll results to test the reliability of the system.

---

> ### Author Response · Authors · 2022-08-15
> **Response to Reviewer awo2**
>
> Thank you for appreciating the value of our work. We hope this dataset could serve as one of the valuable seed questions in the healthcare QA community. Please kindly find our responses below.
>
> **Q: Section 3.2 mentions the concept of "cost sampling procedure". The authors need to provide a more elaborate and clear explanation of how the cost is computed from the MIMIC-III database. Some examples should be used to explain the idea.**
>
> For cost sampling, we take a two-step sampling procedure to simulate the cost values. First, we sample discrete-valued mean costs for four different medical event types (diagnosis, procedure, prescription, and lab events) from a Poisson distribution with a mean of 10. Then, we sample continuous-valued costs from Gaussian distributions with their corresponding means. We have added the details in Section 3.2.
>
> **Q: Even though the paper has experimented with models like T5 and GAP for the EHRSQL dataset, the authors should have also evaluated its performance on the "TREQS model" (TRANSLATE-EDIT MODEL FOR QUESTION-TO-SQL QUERY GENERATION) which was used originally for the MIMICSQL dataset. Or at least, the authors should have mentioned the reason for not evaluating using the TREQS model.**
>
> We have two reasons for excluding TREQS in our experiments: 1) Even though TREQS shows good performance on MIMICSQL, the motivation behind TREQS lies in handling medical abbreviation and condition value parsing and recovering. Different from MIMICSQL, we do not introduce lexical variations in condition values to focus on trustworthy semantic parsing; therefore, we find TREQS not best suitable for our proposed task. 2) According to Bae et al. [1], transfer learning from pre-trained language models outperforms TREQS, which is a shallow attention-based LSTM model, by a large margin on healthcare text-to-SQL tasks. Considering the recent advances in pre-training and transformers, we choose T5 and GAP as more powerful and up-to-date baselines for our text-to-SQL task. To clarify our baseline choice better, we have added the reason for using T5 over TREQS in Section 4.3.
>
> **Reference**
>
> [1] Bae, Seongsu, et al. "Question Answering for Complex Electronic Health Records Database using Unified Encoder-Decoder Architecture." Machine Learning for Health. PMLR, 2021.

---

> > ### Comment · Reviewer_awo2 · 2022-08-29
> > **Thanks**
> >
> > Thank you for answering all my questions.

---

### Official Review · Reviewer_bkd9 · 2022-07-28
**An intreresting paper and a valuable contribution to datasets for QA in health care**

**Rating:** 9
**Confidence:** 3

**Strengths:**

This paper and its associated dataset (EHRSQL) are a significant contribution to NLP for health care. A key strength of EHRSQL is that the seed questions used to build the dataset (1,742 utterances) were collected through consultation with clinical professionals and other stakeholders in the health care system. Another strength of this dataset is that the authors incorporated different types of time granularity in the questions, which is of particular importance for QA in health care. Finally, another strength of this dataset is that the authors included unanswerable questions in the dataset with the purpose of testing the trustworthiness of QA systems and developed a benchmarking task to test the trustworthiness of semantic parsing.

**Weaknesses:**

I could't identify significant weaknesses in this paper.

**Additional Feedback:**

No additional feedback.

**Clarity:**

The paper is well written and the methodology for the data collection and benchmarking is clearly explained.

**Correctness:**

The dataset is designed with a sound methodology, which is especially suitable for the domain of this dataset (health care). The proposed evaluations for the task on trustworthy semantic parsing are adequate for the scope of this paper.


**Documentation:**

The documentation is adequate.

**Ethics:**

There are no significant ethical concerns regarding this dataset. The authors based their work on pre-existing de-identified datasets. As a  further precaution, they added random shuffling in some of the values of the original datasets to ensure the anonymity of the patients.

**Relation To Prior Work:**

The authors acknowledge prior work and they clearly explain how their work is an advancement compared to previous contributions.

**Summary And Contributions:**

This paper presents EHRSQL, a new text-to-SQL dataset for question answering (QA) from electronic health records (EHRs). The paper provides a comprhensive description of the methodology used to create the dataset. In addition, the paper proposes a new semantic parsing task ("trustworthy emantic parsing") and provides benchmark results on this task.

---

> ### Author Response · Authors · 2022-08-15
> **Response to Reviewer bkd9**
>
> Thank you for the positive feedback on our work. In addition to formulating real-world questions in a solvable question-answering task, we also pay particular attention to incorporating time granularity in the question and trustworthiness of QA models, which are particularly important in the healthcare domain.

---

### Author Response · Authors · 2022-08-15
**Thank you for the reviews**

Dear reviewers and ACs,

We thank all the reviewers for their constructive feedback and are pleased that our work has been well-received overall. We have provided responses to each reviewer’s comment and revised our paper according to the suggestions. Please kindly find the responses below.

---

### Comment · Reviewer_foHQ · 2022-08-22
**Ethics review**

No ethics issue. Dataset is collected by conducting a poll explicitly for the purpose of this dataset.

---

### Meta-Review · Area_Chair_ZGUR · 2022-09-13

**Recommendation:** Accept
**Confidence:** 5

**Metareview:**

This work builds a new text-to-SQL dataset, EHRSQL, using two widely studied medical record databases, namely, MIMIC-III and eICU. The questions are collected using a poll conducted with medical experts and hence are realistic (and not synthetic ones). Another interesting aspect of this work is that the questions collected are time-sensitive in nature and the database also includes unanswerable questions from the poll results. This certainly advances the reliability and generalizability of the system.

It would be helpful for the authors to include a discussion o the limitations of this dataset and potential directions one can extend this dataset. The scope of the unanswerable questions is restrictive. Overall, though the work is extremely rigorous, the scope of this research is limited to a relatively restricted setting. Given the specific focus of the work on the healthcare domain, a poster presentation is recommended since the work will be interesting to only a subset of the audience.

---

### Decision · Program_Chairs · 2022-09-16

Accept